# DIAGVULN: A HOLISTIC CONVERSATIONAL BENCHMARK FOR EVALUATING LLMS ON VULNERABILITY ASSESSMENT

## ABSTRACT

With over 20,000 Common Vulnerabilities and Exposures (CVEs) reported annually, software vulnerabilities represent a critical cybersecurity challenge requiring automated assessment tools. While large language models (LLMs) show promise as cybersecurity assistants, existing benchmarks exhibit fundamental limitations: narrow data sources, neglect of contextual information, and focus on single-turn tasks rather than multi-turn analyst workflows. To bridge this gap, we introduce DIAGVULN, the first multi-turn conversational benchmark for LLM-based vulnerability assessment. DIAGVULN comprises 2,000 CVEs across 23 question-and-answer categories, encompassing detection, localization, classification, root cause analysis, exploit reasoning, impact assessment, and patch. We construct high-quality QA pairs in DIAGVULN using retrieval-augmented generation (RAG) based on data collected from diverse sources, validated through LLM-as-a-Judge and conformal prediction based on human expert annotations. Evaluation of five state-of-the-art LLMs using DIAGVULN reveals substantial limitations, with low top-1 accuracy (below 60%) in vulnerable code detection and CVE identification. Our evaluation also demonstrates that current models lack critical reasoning capabilities for reliable vulnerability assessment. DIAGVULN provides a valuable resource for advancing research in evaluating and fine-tuning LLMs for vulnerability assessment.

## 1 INTRODUCTION

Software vulnerabilities (Prodaft, 2023; Google Cloud, 2023) are flaws or weaknesses in a system that attackers can exploit to compromise its confidentiality, integrity, or availability, potentially leading to crashes, data loss, or security breaches. In the past three years, more than 20,000 CVEs (Common Vulnerabilities and Exposures) have been reported annually, reflecting the increasing scale of threats. This surge in vulnerabilities, compounded by a significant shortage of cybersecurity professionals, intensifies the challenge. According to recent reports (Git, 2022), 71% of security teams are struggling to manage the overwhelming workload of vulnerability assessments. Consequently, there is an urgent need to develop effective tools that can assist and support the security analysis of vulnerabilities.

Prior works based on static analysis and predefined patterns often produce high false positives/negatives and fail to generalize to new vulnerabilities (Lipp et al., 2022; Shoshitaishvili et al., 2016). Deep learning methods (Chakraborty et al., 2021; Cheng et al., 2021) improved detection but remain limited to binary classification, without identifying vulnerability types or locations. Recently, LLMs have been applied to vulnerability analysis (Zhou et al., 2024; Zhang et al., 2024; Lekssays et al., 2025; Shimmi et al., 2024), but they mainly address narrow tasks like classification or code localization, offering little support for root cause analysis or fix recommendations. Additionally, existing vulnerability benchmarks (Chen et al., 2023; Fan et al., 2020; Bhandari et al., 2021; Nikitopoulos et al., 2021; Ding et al., 2024) typically rely on single sources, such as public vulnerability databases or fix commits, and therefore lack crucial contextual attributes, including root cause explanations, fix suggestions, and exploit details, which are needed for comprehensive assessment (see Table 1). Recent work (Ruan et al., 2024) adds contextual information about CVEs but omits key sources like discussion forums, limiting its usefulness for fine-tuning and evaluating LLMs. Prior datasets either (i) focus solely on code-level vulnerability classification (Zhou et al., 2019b), (ii) cover only CVE text without associated code (Bhandari et al., 2021), or (iii) provide no

Table 1: Comparison of our benchmark dataset (DIAGVULN) and existing ones. "CVE" represents Common Vulnerabilities and Exposures; "CID" represents commit ID, "Func" represents vulnerable function, "CWE" represents Common Weakness Enumeration, "FL" represents Fault Location, "CVSS" represents Common Vulnerability Scoring System, "Sev" represents Severity, "CPE" represents Common Platform Enumeration, "Ven" represents Vendor, "Pro" represents Product, "EID" represents Exploit ID, "PoC" represents Proof of Concept.

| - | Vulnerability Type | | | Weakness | | Severity | | Vulnerable Product | | | Exploit | | Patch | |
|---|---|---|---|---|---|---|---|---|---|---|---|---|---|---|
| | CVE | CID | Func | CWE | FL | CVSS | Sev | CPE | Ven | Pro | EID | PoC | Text | Code |
| DiverseVul (Chen et al., 2023) | ◐ | ● | ● | ◐ | ○ | ◐ | ○ | ○ | ○ | ○ | ○ | ○ | ○ | ○ |
| Big-Vul (Fan et al., 2020) | ● | ● | ● | ● | ○ | ● | ○ | ○ | ○ | ○ | ○ | ○ | ● | ● |
| CVEfixes (Bhandari et al., 2021) | ● | ◐ | ◐ | ◐ | ○ | ● | ○ | ● | ● | ● | ◐ | ○ | ● | ● |
| CrossVul (Nikitopoulos et al., 2021) | ○ | ○ | ● | ● | ○ | ○ | ○ | ○ | ○ | ○ | ○ | ○ | ○ | ○ |
| PrimeVul (Ding et al., 2024) | ○ | ● | ● | ◐ | ○ | ○ | ○ | ○ | ○ | ○ | ○ | ○ | ○ | ○ |
| Our dataset (DIAGVULN) | ● | ● | ● | ● | ● | ● | ● | ● | ● | ● | ● | ● | ● | ● |

reasoning-oriented annotations (Ding et al., 2024). None of them support multi-step evaluation of how and why an LLM identifies a vulnerability, nor do they allow consistent comparison across models. Detailed comparison with existing datasets is presented in Appendix A. This is because building a comprehensive vulnerability benchmark is challenging since vulnerability information is scattered across many sources and often appears in unstructured or inconsistent forms.

**Our goal.** To address this gap, we introduce a holistic conversational benchmark for vulnerability assessment that integrates multiple vulnerability data sources and supports multiple tasks across the security analyst's workflow. Specifically, we aggregate data from 13 sources and curate 23 key questions spanning tasks such as vulnerability detection, localization, classification, root cause explanation, exploit reasoning, impact assessment, and fix recommendation. In this work, we focus on Linux kernel vulnerabilities, as Linux powers over 96% of top web servers (Volico, 2024) and offers a uniquely rich ecosystem of source code, patches, exploits, and vulnerability data (Thakur, 2025). Our curation pipeline is fully automated using an retrieval-augmented generation (RAG)-based system to generate Q&A pairs, with an LLM-as-a-judge framework to ensure quality, making the approach easily extensible to other domains (e.g., Windows, ICS, IoT).

**Dataset Curation and Validation Pipeline.** The dataset curation process leverages RAG for automated QA generation and uses an LLM as a judge to assess the quality of the generated answers, which contains three stages: (i) *Data Collection and Portfolio Construction:* Collecting vulnerability-related data, including CVE descriptions, patches, exploit code, and relevant discussions. We have selected authentic and trustworthy websites that are reputed among the research community for vulnerability analysis (Chen et al., 2023; Nikitopoulos et al., 2021; Sun et al., 2024). The collected data is then aggregated into a structured format known as the *Vulnerability Portfolio*, which captures all attributes of a vulnerability. (ii) *QA Pair Curation:* Employing a RAG-based system to automatically generate/extract answers to predefined questions using the constructed Portfolio. (iii) *QA Ground Truth Validation:* Applying an LLM-as-a-judge approach to assess the correctness, comprehensiveness, and factual consistency of the QA pairs.

**DIAGVULN Dataset.** Using the presented pipeline, we generated portfolios for 6,342 CVEs. Moreover, we produce 46,000 QA pairs for 2,000 CVEs. We have picked 100 CVEs and performed human label evaluation to calibrate LLM-Judge using conformal prediction (Shafer & Vovk, 2008). This helped us further in identifying the correctness of the QA pairs generated by our pipeline. Doing this procedure helped us to replace the human reviewers for 200 CVEs with a Confidence of 90%.

**Benchmarking State-of-the-Art LLMs on DIAGVULN.** To understand the existing LLMs' capabilities as potential cybersecurity assistants, we conduct a systematic evaluation in three key areas (vulnerability detection, CVE identification and question answering about the identified vulnerability) using five representative LLMs on 200 CVEs from DIAGVULN. Our selection includes both proprietary models (GPT-4o, Claude 4, and Gemini 2.5) and open-source models (Llama 3.3 and Qwen 3). Our final evaluation shows that (i) SOTA LLMs can only achieve 35-55% weighted F1-score in vulnerability detection; (ii) given the vulnerability, LLMs only identify the correct CVE IDs 50-61% of the time with web search enabled; and (iii) while LLMs do well on surface-level attributes (e.g., Function Name), they struggle on reasoning-heavy ones such as Mitigation, Exploit Explanation, and Patch Code.

**Contributions.** Our main contributions are summarized as follows:

- We propose an automatic data curation pipeline for constructing a conversational vulnerability dataset across different sources that cover multiple CVE attributes.
- We combine LLM-as-a-Judge with conformal prediction using a few human annotations for calibration, enabling reliable and scalable validation of the automatically generated QA data.
- We introduce the first dialogue-based vulnerability assessment benchmark dataset (DIAGVULN), supporting a variety of tasks such as detection, localization, root cause explanation, exploitation, security impact assessment, and fix suggestions.
- We comprehensively evaluate the performance of five state-of-the-art (SOTA) LLMs (both commercial and open-source) on our benchmark dataset to assess their capabilities in vulnerability assessment and reasoning.

## 2 RELATED WORK

**Cybersecurity benchmark for evaluating LLMs.** Recent efforts have sought to establish benchmarks for evaluating LLMs in the cybersecurity domain. A dominant thread curates multiple-choice question (MCQ) datasets to assess factual security knowledge. SecEval (Li et al., 2023) introduces 2,000 MCQs across domains such as software, application, and system security, while in CyberMetric (Tihanyi et al., 2024) expands coverage to 10,000 MCQs spanning cryptography, reverse engineering, and risk assessment, with expert validation. SecQA (Liu, 2023) similarly derives MCQs from a security textbook to probe foundational principles. Moving beyond recognition, SecBench (Jing et al., 2024) augments 44,823 MCQs with 3,087 short-answer questions across nine sub-domains, enabling evaluation of both knowledge retention and reasoning. Other benchmarks emphasize task-specific evaluations, such as CTIBench (Alam et al., 2024) for cyber-threat intelligence (e.g., CVE–CWE mapping, CVSS scoring, ATT&CK extraction) and SECURE (Bhusal et al., 2024) which targets industrial control systems through six datasets grounded in industry standards. Different from these studies, our work curates the first multi-turn conversational benchmark for vulnerability assessment in the Linux domain, designed to evaluate LLMs across a broader spectrum of tasks (e.g., vulnerability localization and questions on severity, patches, exploits, and impacts) for a more comprehensive assessment of their capabilities as security assistants.

**LLM for vulnerability analysis.** While LLMs show promise for vulnerability analysis, recent studies reveal that off-the-shelf models often struggle with poor reasoning, hallucinations, and non-deterministic behavior. Ullah et al. (2024) evaluated LLMs on security bugs and found they often fail to localize or reason about key constructs such as bounds checks, NULL checks, and pointer operations. Yin & Ni (2024) employed a Big-Vul (Fan et al., 2020) benchmark to assess LLMs' capabilities, finding that pre-trained models excel in detection, while CodeLlama and WizardCoder perform better in assessment and description with sufficient context. SCoPE (Gonçalves et al., 2024) showed that simple preprocessing, such as removing formatting noise, can lead to modest improvements in detection accuracy. Kouliaridis et al. (2024) evaluated nine LLMs on Android OWASP Mobile Top 10 vulnerabilities, highlighting highly variable performance across domains. Similarly, Lin & Mohaisen (2025) assessed numerous LLMs under different configurations, including model size, quantization, and context window on C/C++ vulnerabilities, finding significant variation in performance across models and languages. Liu et al. (2024a) compared ChatGPT against state-of-the-art vulnerability management tools, revealing strengths in summarization but weaknesses in complex reasoning tasks. Chen et al. (2024) conducted a systematic evaluation on detecting execution-injection bugs (EIBs), reporting that although GPT-4 shows potential, its limited precision and recall hinder its practical applicability. While these studies provide valuable evaluations, they often rely on limited LLM memory or narrow datasets, constraining scalability and task diversity. Our work addresses this gap by automatically curating rich, contextualized and conversational vulnerability data, creating a backbone for comprehensive benchmarks and fine-tuning LLMs to improve reasoning and real-world applicability.

## 3 DATASET CURATION AND VALIDATION PIPELINE

This section describes our dataset curation and validation pipeline for constructing high-quality QA pairs. As shown in Figure 1, the pipeline consists of three components: (1) Data Collection and Portfolio Construction, (2) QA Pair Curation, and (3) QA Ground Truth Validation.

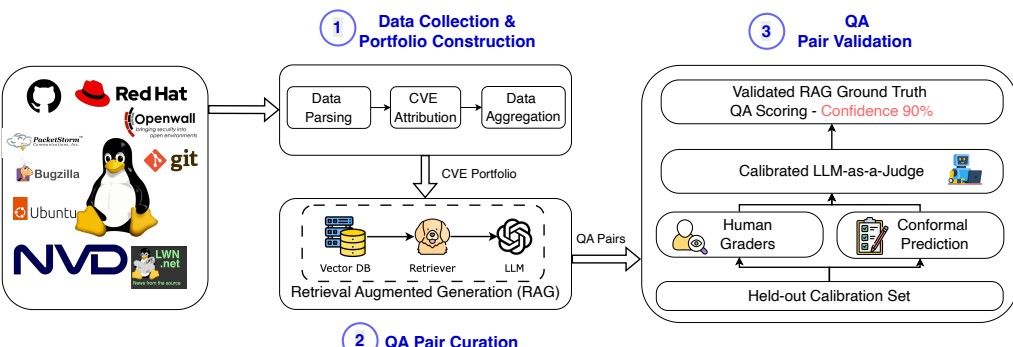

Figure 1: Dataset Curation and Validation Pipeline: (❶) Construction of comprehensive portfolio capturing vulnerability attributes, exploit information, patches, and developer discussions from diverse sources, including community-maintained vulnerability databases, exploit repositories, security advisories, and vulnerability disclosure mailing lists. (❷) Generation of QA pairs that contain descriptive and coding answers using retrieval-augmented generation (RAG). (❸) Validation of RAG-generated QA pairs as ground truth, via LLM-as-a-Judge calibrated with conformal prediction, which (i) assesses answer quality, and (ii) assigns calibrated confidence scores using few human annotations.

### ① Data Collection and Portfolio Construction

Vulnerability information in our dataset is collected from National Vulnerability Disclosure (NVD) database, 7 independent platforms (e.g., GitHub, Ubuntu, Redhat), and 5 public disclosure forums (Seclists, OpenWall, Exploit-DB, Packetstormsecurity and Lkml). Such contextual information is essential for generating high-quality QA pairs that support security practitioners in understanding and addressing vulnerabilities effectively. We have divided our information sources as structured and unstructured data sources. Structured data, organized in consistent formats (e.g., XML), is from repositories such as the National Vulnerability Database (NVD) (NIST, 2024) and Ubuntu Security Notices. On the other hand, unstructured sources involve just the information as a mailing list or a developer discussion (e.g., `seclists.org`, `lkml.org`) and community forums (Red Hat). This multi-source integration strategy enables us to capture a holistic view of each vulnerability, incorporating both technical artifacts and human-driven insights.

Once the data is collected, we organize it into a hierarchical representation termed the *Vulnerability Portfolio*. Each Portfolio captures a multidimensional view of a CVE, encompassing aspects such as exploit code, patch details, severity metrics, and community discussion summaries, which enable systematic access to vulnerability information across multiple dimensions and facilitate downstream QA pair generation. Converting all the downloaded data into portfolio involves three key post processing steps: (i) Data Parsing: This involves extraction of useful information from the gathered structured and unstructured sources. (ii) CVE Attribution: This step involves mapping each data point to appropriate CVE ID. For developer discussions and code comments, we used regular expressions to extract CVEs. (iii) Data Aggregation: we aggregate all parsed attributes into unified "portfolio". Using the NVD CVE list as the reference set, we align entries from other sources and retain only CVEs with sufficient context for downstream QA tasks, prioritizing those enriched with exploit details, developer discussions, patch data, and corroborating evidence. If some fields of a CVE have limited details across the sources, such fields are stored with "Not Found" values in the portfolio. The structure of a portfolio with details about the post-processing are presented in Figure 4 in Appendix B.

### ② QA Pair Curation

Although each portfolio structurally aggregates all information for a CVE, it cannot be directly used for LLM related tasks as it lacks distinctive explanations.

To extract information and curate QA pairs from portfolio, we employ Retrieval-Augmented Generation (RAG). RAG segments documents into chunks, embeds them, retrieves the most relevant fragments for a query, and then prompts an LLM to generate an answer grounded in the extracted fragments. We avoided relying on LLMs for generating the explanations about vulnerabilities. Instead, we use the LLM in the RAG system, where it focuses solely on retrieving factual information. This approach offers three key advantages: (i) QA-style format-

Table 2: **LLM-as-a-Judge Rubric for Scoring Faithfulness, Correctness, and Completeness.** Claim-level scores are averaged and converted to Likert-scale (1−5) for Faithfulness and Correctness; Completeness is judged holistically in (1-5).

| Level | Score | Faithfulness | Correctness | Completeness |
|-------|-------|--------------|-------------|--------------|
| Claim | 1.0 | Explicitly stated or supported. | Factually accurate and logically valid. | N/A |
|       | 0.5 | Plausible but not fully verifiable. | Ambiguous but not contradicted. | N/A |
|       | 0.0 | Unsupported or unverifiable. | Incorrect, contradicted, or irrelevant. | N/A |
| Response | 5 | Average claim score higher than 0.9 | | Fully address the question. |
|          | 4 | Average claim score between 0.7 and 0.9 | | Substantially address the question. |
|          | 3 | Average claim score between 0.4 and 0.7 | | Partially address the question. |
|          | 2 | Average claim score between 0.2 and 0.4 | | Minimally address the question. |
|          | 1 | Average claim score between 0.0 and 0.2 | | No answer to the question. |

ting, (ii) evidence-grounded responses, and (iii) reduced hallucinations. The choice of model for RAG is primarily based on the available flagship reasoning model (GPT-4o). Experiments with other models and implementation details of the RAG pipeline are provided in Appendix F.

In total, we designed 23 user prompts for the attributes to represent each CVE. In addition to the QA prompts generated by the RAG system, we have also included direct attributes such as impact scores from the portfolio. For CVEs with insufficient supporting information, the RAG system is designed to return a "Not Found" placeholder for the corresponding entries in the question–answer pairs. After running the RAG system on each CVE for every attribute, we use these answers with human-like questions as QA pairs. A sample conversational dialogue after generating the QA-pairs is presented in Figure 2. The complete list of questions in QA pairs and the list of all the attributes are presented in Appendix G.

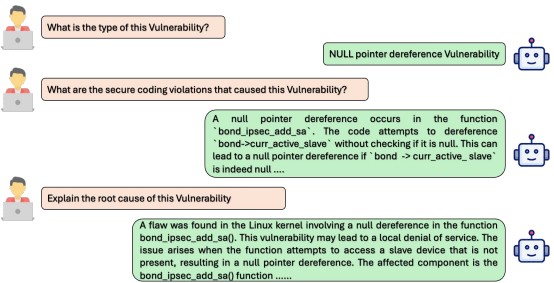

Figure 2: Sample Conversational QA pairs

### ③ QA Ground Truth Validation

(i) *Validation Criteria:* We focus on three quality assessment criteria: Faithfulness, Correctness, and Completeness. These metrics are designed to comprehensively evaluate the quality and reliability of LLM-generated outputs (Li et al., 2024), which are defined as follows:

• Faithfulness: As a widely adopted metric for evaluating RAG-based question answering (Gan et al., 2025; Es et al., 2024), Faithfulness measures how accurately a generated response reflects, and remains grounded in the relevant context (i.e., the CVE portfolio), penalizing any hallucinations.
• Correctness: This criterion checks whether a RAG-generated response is factually accurate and appropriate to the user-specified question. Here, we use the RAG context retrieved specifically for each CVE attribute as a proxy for human-annotated ground truth, since this context contains the factual answer to the question.
• Completeness: This criterion assesses how well the generated response addresses the intent and content of the user question, a quality commonly discussed as relevance (Gan et al., 2025; Es et al., 2024; Rengo et al., 2025). Particularly in our setting, Completeness goes beyond general relevance by requiring that every user-specified aspect of each CVE attribute is addressed.

(ii) *LLM-as-a-Judge:* Inspired by recent works on automated quality assessment of LLM/RAG-generated texts using LLMs with chain-of-thoughts (CoT) as a judge (Liu et al., 2023; Zhu et al., 2024; Es et al., 2024), we design a framework to measure the above three metrics in a fully automated way by using an LLM. Furthermore, we incorporate context-specific interpretation instructions for each attribute, enabling fine-grained judgment aligned with human annotation standards.

Following recent LLM-as-a-Judge frameworks (Es et al., 2024; Zhu et al., 2024), we begin by decomposing each RAG-generated response into a set of discrete factual claims to evaluate both *Faithfulness* and *Correctness*, except for coding-related attributes (e.g., Exploit Code, Patch Code). Each extracted claim is scored independently using the rubric in Table 2 based on: verifiability from retrieved context for *Faithfulness*; factual accuracy and logical consistency for *Correctness*. The average claim-level score is then converted into a 5-point Likert scale, also defined in Table 2, to

Table 3: Different information sources and their type

| Source | Download Count | CVEs available | Download category | Attributes |
|---|---|---|---|---|
| NVD | 14170 | 9379 | All Linux CVEs | Description, References |
| GitHub | 1300 | 987 | Reference links | Git diff messages |
| git.kernel.org | 24233 | 6325 | Complete commit messages | Commit messages and Patch details |
| Seclists | 2789 | 344 | Complete website | Developer discussions |
| Red-hat | 6967 | 6958 | CVE specific | Impact and Description |
| OpenWall | 573 | 386 | Reference links | Developer discussions |
| Exploit-DB | 3095 | 2183 | All Linux exploits | Exploit code |
| bugzilla.kernel.org | 586 | 544 | Reference links | Vulnerability details |
| bugzilla.redhat.com | 5743 | 5743 | CVE specific | Description |
| Lkml | 154 | 118 | Reference links | Developer discussions |
| Packetstormsecurity | 83 | 63 | Reference links | Exploit description and Code |
| lwn | 324 | 154 | Complete website | Vulnerability description |
| Ubuntu | 8106 | 8106 | CVE specific | CVE metrics and Description |

produce the final response-level score. Here, we adopt this score conversion because Likert-scale scores offer more intuitive human interpretation (Robinson & Leonard, 2019), align with recent LLM evaluation practice (Liu et al., 2023; Zhu et al., 2024), and facilitate subsequent conformal prediction with human judgments. Furthermore, since *Completeness* requires holistic judgment of the entire response, we directly apply the 5-point rubric (Table 2) to evaluate how thoroughly the RAG-generated responses address the user-specified questions about CVE attributes. The prompt used for evaluation in LLM-Judge is presented in Appendix L.

(iii) *Conformal Prediction:* We adopt conformal prediction to validate the reliability of LLM-as-a-Judge scores for RAG-generated outputs, which will serve as the ground-truth labels. Conformal prediction Shafer & Vovk (2008); Angelopoulos & Bates (2021) is a *model-agnostic and distribution-free* uncertainty quantification framework that outputs a prediction interval or set that is guaranteed to contain the true value with probability of at least $1 - \alpha$ given a user-specified significance level $\alpha$ (e.g. $\alpha = 0.1$). Different from previous works that employ conformal prediction (Mohri & Hashimoto, 2024; Wang et al., 2024; Su et al., 2024; Ye et al., 2024) for correctness-controlled generation or uncertainty scoring, we adopt it to estimate how closely the automatic scores returned by an LLM evaluator (e.g., GPT-4.1) match human annotations. Specifically, we compute the absolute differences between human-assigned and LLM-assigned scores on a **held-out calibration set**, then select a deviation threshold as a high $(1 - \alpha)$-th quantile of those differences (e.g., 90th percentile). Subsequently, we use this threshold to establish high-confidence bounds on the expected difference between LLM scores and human labels. More details about Conformal Prediction are presented in Appendix H.

## 4 DiagVuln Dataset

① **DiagVuln Creation.** Using the methodology discussed in section 3, we align all collected artifacts of individual CVEs and curate a compact set of high-value sources based on coverage, artifact richness, and relevance. Structured repositories such as NVD and Ubuntu provide canonical identifiers and metadata. Official Git histories (e.g., `git.kernel.org`) supply patches and vulnearble function details. Exploit archives add concrete attack artifacts, while developer discussions (e.g., Seclists, OpenWall, `lkml.org`) capture rationale and impact. The result is a unified, context-rich CVE representation designed for downstream analysis. Table 3 summarizes the sources and CVE counts, with a detailed breakdown provided in Appendix C. As our goal is to support realistic security analyst workflows, where security professionals use LLMs to assist in vulnerability assessment by asking questions such as *"What is the root cause?"*, we curated portfolios for about 6,342 CVEs. Then, we generated 40,000+ QA pairs for 2,000 recent CVEs, constrained by the cost of OpenAI API tokens. Our pipeline, however, is generalizable and can be extended to the full set of CVEs. Out of the 2,000 CVEs, 80% of the CVEs can be used to fine-tune an LLM and 20% of the CVEs can be used as test set.

② **Human Assessment.** The aim of performing human evaluation is to verify the generated results of the RAG system and label the ground truth across different attributes. We have selected QA pairs of 100 CVEs at random and these are picked randomly from the available set of CVEs based on rich contextual information across attributes. The diversity of these attributes is presented in Appendix K. The QA pairs were then validated by two of the annotators with domain knowledge about Vulnerabilities, with a third author to resolve any conflicts. Across 100 CVEs, this process

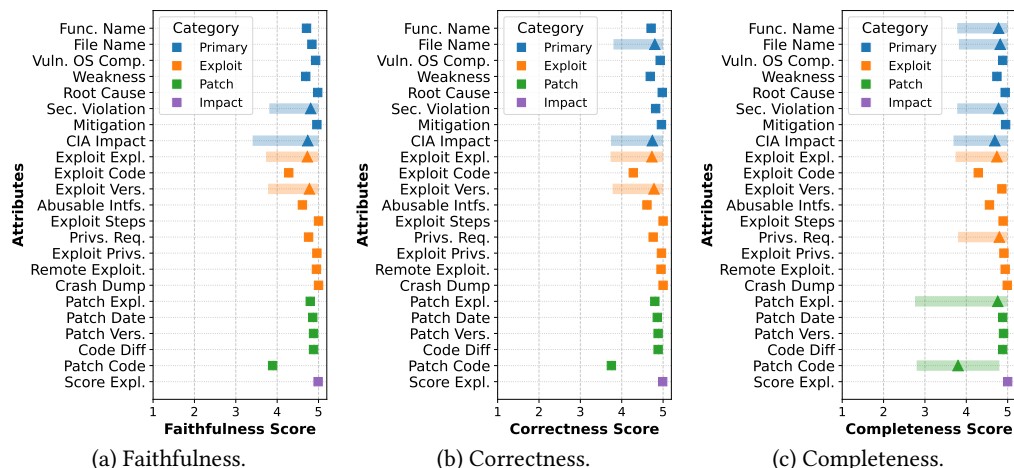

(a) Faithfulness.    (b) Correctness.    (c) Completeness.

Figure 3: **Conformal-Calibrated Scores of RAG-generated CVE Attributes.** Average LLM-judged (a) *Faithfulness*, (b) *Correctness*, and (c) *Completeness* scores for 23 CVE attributes are illustrated as markers, with 90% conformal confidence intervals shown as bars. Square markers (■) indicate perfect agreement (zero non-conformity) with human scores, while triangles (▲) mark attributes with LLM–human deviations. The figure highlights that the RAG-generated attributes not only achieve high evaluation scores, most of them also exhibit strong LLM-human score agreement under 90% conformal prediction confidence.

required about 50 hours per annotator, with an additional 10 hours for conflict resolution. We have converted all the descriptive responses from the RAG system into readable claims (similar to LLM-Judge). For each of these claims, the subjects are required to answer claim level detail based on *Faithfulness* and *Correctness*. The RAG answer is also holistically measured using *Completeness* metric. Further details about the human labeling is presented in Appendix I.

③ **Validation of RAG using LLM-as-a-Judge.** Validating the quality of RAG-generated QA pairs ideally requires comparison with human-annotated ground truths. Yet creating expert-reviewed 46,000 QA pairs of more than 2,000 CVEs is infeasible. To achieve this, we adopt an LLM-as-a-Judge framework to automatically score QA pairs without annotations, and we apply conformal prediction to calibrate the confidence of these judgments. Figure 3 summarizes the average LLM-judged Faithfulness, Correctness, and Completeness scores across 23 QA attributes, averaged over 200 CVEs randomly selected from our 2,000-CVE dataset. The human annotation scores are presented in Appendix S. Most attributes achieve near-perfect scores (close to 5), supported by 90% conformal confidence intervals that closely match human judgments. This strong performance reflects the high-quality, attribute-specific contexts retrieved from CVE portfolios. The main exception is the Patch Code attribute, which shows lower completeness and wider confidence intervals due to occasional retrieval of incomplete or superficial code snippets (e.g., formatting changes or unrelated metadata). These issues highlight opportunities for improving patch curation in future iterations of the RAG system.

Overall, these results validate the reliability of LLM-as-a-Judge for scoring RAG-generated QA pairs in the absence of human annotations. Furthermore, given the high average scores and conformal agreement guarantee, we confirm that the RAG-generated QA pairs are sufficiently accurate to serve as reference soft labels for evaluating other LLMs in subsequent experiments. In the following section, we will use a **different** LLM-based judge (i.e., distinct from the soft label validator), to assess the Correctness of SOTA LLM outputs relative to the RAG soft labels.

## 5 LLM PERFORMANCE EVALUATION

In this section, we conduct extensive experiments to evaluate the performance of five representative LLMs: GPT-4o (Hurst et al., 2024), Claude Sonnet 4 (Anthropic, 2025a), Gemini 2.5 Pro (Comanici et al., 2025), Qwen3 (Yang et al., 2025), and Llama 3.3 (Dubey et al., 2024) on DIAGVULN.

**Evaluation Dataset.** To efficiently evaluate five SOTA LLMs, we select a 10% subset of DIAGVULN consisting of 200 CVEs. The subset was selected to be representative, spanning diverse vulnerability types and years, and to include the richest contextual information (e.g., patches, exploits,

discussions) for effectively testing LLM reasoning. Specifically, we focus on 200 CVEs with complete vulnerable code snippets and validated RAG ground truth. From these CVEs, we extract 83 non-vulnerable code patches under 128K tokens to fit the context limit of open-source models.

**Evaluation Procedure.** We evaluate each LLM model using a 3-step procedure representative of practical vulnerability assessment workflow, which are described as follows:
• Step 1 – *Vulnerability Detection:* We only input the file names and corresponding code snippets to the LLMs, and task them to detect code vulnerability as binary classification.
• Step 2 – *CVE Identification:* As CVE IDs are directly related to trusted documentations of the vulnerability, investigating LLMs' knowledge on CVE identification is crucial. Here, LLMs receive file names and code snippets with the expected vulnerable behavior. They must infer the correct CVE identifier and provide a brief explanation for the match.
• Step 3 – *CVE Attribute Question-Answering:* Afterwards, the correctly identified CVEs in the previous step, along with their file names and code snippets, are given to LLMs. Then, the LLMs will answer a series of fine-grained questions about the vulnerability.

**Prompting Strategy.** We utilize a single model-agnostic system prompt to ensure neutrality and consistency, then attach task-specific user prompts for each evaluation step. For the CVE Identification (step 2) and CVE Attribute QA (step 3), which require external CVE knowledge, we incorporate a unified web-search pipeline implemented with GPT-4o. This standardization removes variability in web-search capabilities across LLMs, while isolating their reasoning ability for the evaluation. Firstly, GPT-4o composes context-aware search queries for each sample. Then, its web browsing tool processes the queries to retrieve the top 5 results from online CVE sources. Finally, the retrieved results will be injected into the prompts for steps 2 and 3. Full prompt templates are detailed in Appendix M.

**Evaluation Metrics.** We adopt task-specific scoring metrics to assess model performance across the three vulnerability assessment steps. For Vulnerability Detection, we use *Weighted* Precision, Recall, and F1-Score due to class imbalance in the testing dataset (200 *vulnerable* vs. 83 *non-vulnerable* samples). Regarding CVE Identification, given a ranked list of top-5 CVE IDs for each sample returned by LLMs, we calculate the Top-1 and Top-5 prediction accuracies. Lastly for CVE Attribute Question-Answering, the LLMs responses to 23 fine-grained questions pertaining to vulnerability attributes (e.g., root cause, exploit mechanism, mitigation strategy, etc.). We compare each LLM-generated response against the corresponding trusted RAG-generated soft label, scoring Correctness on the 1-5 scale using an LLM-based judge. Full details on the evaluation metrics are available in Appendix J.

**Evaluation Results.**
(i) *Vulnerability Detection Effectiveness.* The results in Table 4 demonstrate that all models exhibit poor performances in distinguishing vulnerable and non-vulnerable test samples. Among the five evaluated LLMs, Gemini 2.5 achieves the highest weighted F1 score (55.25%), followed by Claude 4 (53.95%). Additionally, the substantial discrepancy between weighted Precision and Recall across most models is primarily attributed to two factors: (i) the skewed class distribution in the test set (200 vulnerable vs. 83 non-vulnerable samples), and (ii) the inherent classification biases of the LLMs. For example, Gemini 2.5 exhibits a strong bias toward the vulnerable class. It correctly identifies 178 out of 200 truly vulnerable samples, but misclassifies 82 out of 83 non-vulnerable ones. In contrast, Llama 3.3 displays the opposite behavior, strongly favoring the non-vulnerable class. While this conservative behavior reduces false positives, it fails to detect real threats (35.12% weighted F1), rendering the model ineffective in security-critical settings.

(ii) *CVE Identification Effectiveness.* We also investigate the LLM's capability to retrieve the correct CVE identifiers given vulnerable code. Table 5 summarizes the Top-1 and Top-5 CVE identification accuracies across five LLMs, evaluated under two conditions: with and without web search integration (data provided by GPT-4o web search). Without web search, most models fail to retrieve or match real CVE identifiers based on their internal knowledge. On the other hand, with web search integration, all models demonstrate substantial performance gains, where Gemini 2.5 achieves the best performance, with 60.50% Top-1 and 78.00% Top-5 accuracy. These improvements confirm that access to online vulnerability documentations is critical for accurate CVE identification.

(iii) *CVE Attribute Question-Answering Effectiveness.* Finally, we assess how well current LLMs can answer fine-grained questions about specific CVE attributes, compared to trusted RAG-generated

Table 4: **Vulnerability Detection Effectiveness.** We report the *weighted* Precision, Recall, and F1 scores in percentages, with the best results in **bold**. From the table, Gemini 2.5 has the best performance. However, the low weighted F1 scores highlight that existing LLMs cannot detect vulnerability from code snippets with high accuracy.

| LLM Models | w-Precision | w-Recall | w-F1 |
|---|---|---|---|
| GPT-4o | 58.54 | 51.24 | 53.40 |
| Claude 4 | 54.19 | 53.71 | 53.95 |
| Gemini 2.5 | 49.66 | **63.25** | **55.25** |
| Qwen 3 | 56.51 | 44.17 | 46.05 |
| Llama 3.3 | **69.93** | 40.07 | 35.12 |

Table 5: **CVE Identification Effectiveness.** We report the Top-1 and Top-5 accuracy scores in percentages, with (✔) and without (✘) web search (WS). Best results are highlighted in **bold**. We can observe that existing LLMs still struggle to link known vulnerabilities with the correct CVE IDs, despite web search substantially boosting their accuracies.

| LLM Models | (✘WS) Top-1/5 Acc. | (✔WS) Top-1/5 Acc. |
|---|---|---|
| GPT-4o | 5.00 / 9.00 | 55.50 / 74.50 |
| Claude 4 | **17.20 / 29.00** | 56.50 / 76.50 |
| Gemini 2.5 | 5.50 / 8.50 | **60.50 / 78.00** |
| Qwen 3 | 0.00 / 0.00 | 56.50 / 75.00 |
| Llama 3.3 | 1.51 / 3.52 | 50.00 / 72.56 |

Table 6: **CVE Attribute Question-Answering Effectiveness.** Correctness scores (1–5) of LLM outputs, judged against RAG references. Best results are in **bold**, second best are underlined. Claude 4 overall yields closest responses to RAG. While LLMs do well on surface-level attributes (e.g., Function Name, Remote Exploitability), they struggle on reasoning-heavy ones such as Mitigation, Exploit Explanation, and Patch Code.

| Category | Attribute | GPT-4o | Claude 4 | Gemini 2.5 | Qwen 3 | Llama 3.3 |
|---|---|---|---|---|---|---|
| Primary | Function Name | 3.67 | 3.16 | 3.45 | 3.42 | **3.74** |
| | File Name | **3.85** | 3.53 | 3.69 | 3.20 | 3.78 |
| | Vulnerable OS Components | 3.96 | **4.50** | 4.35 | 4.40 | 4.07 |
| | Weakness Type | 3.02 | **3.30** | 2.84 | 2.99 | 3.31 |
| | Root Cause | 4.22 | **4.65** | 4.50 | 4.14 | 4.42 |
| | Secure Coding Violations | 3.57 | 4.06 | **4.12** | 3.91 | 3.10 |
| | Mitigation | 2.11 | 2.05 | **2.30** | 2.15 | 1.91 |
| | CIA Impact | 3.92 | **4.32** | 4.19 | 3.84 | 4.12 |
| Exploit | Exploit Explanation | 2.69 | **3.00** | 2.89 | 2.65 | 2.77 |
| | Exploit Code | **3.70** | 3.02 | 2.31 | 2.77 | 3.62 |
| | Exploited Versions | **2.94** | 2.59 | 2.66 | 2.61 | 2.57 |
| | Abusable Interfaces | 3.61 | 3.06 | **3.67** | 3.65 | 3.19 |
| | Exploit Steps | 3.40 | **3.51** | 3.38 | 3.11 | 3.28 |
| | Privileges Required | 3.77 | **4.33** | 4.17 | 4.17 | 3.49 |
| | Exploit Privileges | 3.03 | **4.34** | 4.04 | 3.32 | 2.65 |
| | Remote Exploitability | 4.39 | 4.56 | **4.70** | 4.04 | 4.48 |
| | Crash Dump | 3.09 | 1.74 | 2.87 | 3.00 | **3.88** |
| Patch | Patch Explanation | 3.50 | **3.89** | 3.81 | 3.34 | 3.38 |
| | Patch Date | 1.31 | 1.52 | 1.26 | 1.12 | **1.88** |
| | Patched Versions | 1.84 | 1.79 | **1.89** | 1.70 | 1.64 |
| | Code Difference | **2.53** | 1.83 | 1.69 | 1.73 | 2.50 |
| | Patch Code | 1.50 | 2.54 | **2.65** | 2.22 | 1.28 |
| Impact | Score Explanations | 2.78 | **3.79** | 3.28 | 2.71 | 3.25 |

references. Here, we focus on successfully identified CVEs in step 2 for detailed question-answering, because LLM models already have access to the correct CVE context from web search. Table 6 presents the Correctness scores across 23 attributes judged by an LLM, which compares the RAG-generated answers with those of SOTA LLMs. Among the evaluated models, Claude 4 performs the most consistently across all attributes, achieving the closest match to RAG-generated answers and ranking first on 10/23 attributes. This highlights its strength in both reasoning about specific vulnerability aspects and leveraging unstructured information retrieved via web search.

Additionally, all LLMs score approximately 4.0 or higher in *Correctness* on surface-level attributes such as Root Cause, Vulnerable OS Component, and Remote Exploitability, which can be easily extracted from the context. For structural attributes like Exploit Code and Crash Dump, GPT-4o and Llama 3.3 also demonstrate strong performance, showcasing their ability to extract well-defined patterns from code snippets. In contrast, reasoning-intensive attributes including Mitigation, Exploit Explanation, and Patch Code pose consistent challenges across all models. For instance, Mitigation receives the lowest score among the Primary attributes for every model, suggesting that LLMs lack the capability to synthesize nuanced preventive strategies. Likewise, attributes under the Patch category generally yield the lowest Correctness scores. This underscores the difficulty in locating specific contextual details like Patch Date and Patched Versions, or reasoning over in-

complete/noisy code changes for Patch Code. We have picked 12 CVEs with correctness score less than 3 for "mitigation" as a sample experiment presented our analysis of the result in Appendix K.

These findings emphasize that while current LLMs are capable of extracting explicit facts from well-structured input, they remain limited in performing technical reasoning over complex, fragmented, or implicit evidence in the zero-shot setting. To address this gap, exploring LLM fine-tuning with attribute-specific QA examples, such as those in our DiagVuln dataset, is necessary to improve the reliability of LLMs on high-stake software vulnerability assessment tasks.

To check whether fine-tuning based on our dataset can improve the scores of LLMs, we conducted a 3-shot in-context fine tuning using two SOTA LLMs and found that the performance of LLMs were improved. The results are presented in Appendix P.

## 6 Discussion

**Optimized prompts to evaluate SOTA LLMs.** We additionally evaluated Claude 4 and Llama 3.3 using model-specific optimized prompts for the CVE attribute QA task, with results summarized in Appendix O. Optimized prompts slightly improve Correctness on 12/23 attributes for Claude 4 and 14/23 for Llama 3.3, while the remaining attributes show similar or slightly reduced performance. This modest improvement is expected, as our model-agnostic prompt already incorporates several prompt-robustness strategies (Anagnostidis & Bulian, 2024), including strict output formats and step-by-step instructions for technical attributes. The optimized variants further constrain Claude 4 to concise answers (Anthropic, 2025b) and impose tighter context and reasoning controls on Llama 3.3 (Meta Platforms, 2025). The results in Appendix O indicate that optimized prompts offer limited gains, suggesting that our evaluation in Section 5 is robust to prompt variations.

**Detection and Identification Evaluation Phases.** We evaluated the false positives from phase 1 with phase 2 and also evaluated the system by combining these two phases. When passed all false positives from the detection phase into the identification phase inevitably leads the model to generate incorrect CVE IDs (hallucinations). In a combined setting, when non-vulnerable samples incorrectly flagged as vulnerable propagate into the identification step, the model produces substantially more incorrect CVE predictions, resulting in reduced overall identification accuracy relative to the decoupled setup. The results are presented in Appendix R.

**Potential Risk of Data Leakage.** LLMs are pretrained on large corpora of text and code, which may include blog posts or code patches related to some of the vulnerabilities in our evaluation. To probe this possibility, we evaluate GPT-4o on CVE attribute question-answering (QA) for 10 less-documented CVEs. As shown in Table 14 (Appendix Q), the Correctness scores slightly decrease on 16/23 QA attributes for these less-documented CVEs. This reduction can be explained either by the weaker documentation footprint of these CVEs or by limited leakage on the original 200 CVEs with richer context. Additionally, our CVE identification results in Table 5 indicate that most LLMs have very limited specialized knowledge about our 200 test samples, for instance GPT-4o correctly identifies only 10 CVE IDs out of 200 samples without searching for online information. Therefore, although we cannot directly measure the extent of possible data leakage, the poor CVE identification performance and the modest drop in attribute-level QA on less-documented CVEs suggest that any data leakage, if present, is rare and minor.

## 7 Conclusion and Future Work

In this work, we presented DiagVuln, the first conversational benchmark for vulnerability assessment that integrates 13 data sources and supports 23 task dimensions. Our automated RAG-based pipeline, validated with an LLM-as-a-Judge and conformal calibration, enables scalable and reliable QA curation. Evaluating five SOTA LLMs on DiagVuln, reveals strong performance on surface-level tasks but persistent gaps in reasoning-heavy ones, highlighting the need for more advanced models. Looking forward, we plan to use DiagVuln, to fine-tune domain-specific LLMs that can better reason about vulnerabilities and provide actionable guidance. We also aim to extend the benchmark to other domains such as Android and critical infrastructure platforms (e.g., Industrial Control Systems), broadening its impact across diverse security ecosystems.

ETHICS STATEMENT

The authors have read and fully complied with ICLR Code of Ethics. Our data was collected from public sources. Moreover, we sent our scraping requests at a large interval (e.g., 5 minutes) to the websites in order to prevent any unintentional interference to the websites. Therefore, there should be no ethical concerns.

REPRODUCIBILITY STATEMENT

Our data collection pipeline, DIAGVULN dataset, and the evaluation results of SOTA LLMs are available at: https://anonymous.4open.science/r/rag_llm_vuln-3F6E. To ensure reproducibility, we have included all our scripts, along with documentations (e.g., READMEs) and instructions for running the code.

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

## A    Further details on related work and motivation

The Table 1 of the paper details the previous datasets and how our proposed dataset is different from the previous works. The detailed comparison with the existing vulnerability detection and assessment datasets are as follows:

(1) Vulnerability data sources There is no comprehensive, fine-grained benchmark that systematically evaluates LLM performance across the full vulnerability-assessment workflow, from detection to attribution, localization, patch explanation, and reasoning. Prior datasets either (i) focus solely on code-level vulnerability classification (Zhou et al., 2019a), (ii) cover only CVE text without associated code (Bhandari et al., 2021), or (iii) provide no reasoning-oriented annotations (Ding et al., 2024). None of them support multi-step evaluation of how and why an LLM identifies a vulnerability, nor do they allow consistent comparison across models.

(2) Dataset dimensionality: Additionally, prior attempts to build cybersecurity datasets frequently suffer from small data volume or rely on multiple-choice question formats, which constrain task diversity and limit their usefulness for instruction tuning. These evaluations also depend on narrow datasets or short LLM memory, restricting scalability and preventing support for multi-turn conversations.

(3) Data quality and correctness: While there are other CVE datasets that were presented (Bhandari et al., 2021; Ding et al., 2024; Fan et al., 2020; Nikitopoulos et al., 2021; Ruan et al., 2024; Chen et al., 2023), most of them neither propose a concrete evaluation procedure nor have a scalable approach for evaluation even when the dataset is labeled using automated techniques. DiverseVul (Chen et al., 2023) contains direct crawled data from internet and acknowledged that the proposed dataset might have a limitation of label noise. In addition to (Chen et al., 2023), other datasets (Bhandari et al., 2021; Fan et al., 2020; Nikitopoulos et al., 2021) were also shown to be containing mislabeled vulnerable samples (Ding et al., 2024). PrimeVul (Ding et al., 2024) offers the most systematic evaluation to date, demonstrating substantial reductions in label noise across previously released datasets, including its own. However, its evaluation process relies heavily on manual inspection and therefore does not scale to larger corpora. The latest work (Ruan et al., 2024) provides an automated and scalable data collection pipeline but did not have a concrete evaluation strategy to assess the quality of the dataset.

In response to these gaps, we introduce a dataset that combines scalability with a clearly articulated evaluation protocol. Our assessment framework employs both human review and LLM-as-a-Judge strategy to systematically measure and validate labeling accuracy. Even though our manual evaluation of 100 CVEs with 3 human subjects took around 60 human hours, this can be used in conjunction with LLM-as-a-Judge for scaling up to further new samples.

(4) Suited for LLM training and evaluation: A series of recent evaluations— Purple Llama CyberSecEval (Bhatt et al., 2023), CyberSecEval 2 (Bhatt et al., 2024), and CyberSecEval 3 (Wan et al., 2024)—focus on assessing LLM behavior in security-sensitive contexts such as insecure code generation, cyberattack assistance, interpreter abuse, prompt injection, and broader offensive capabilities. CyberBench (Liu et al., 2024b) further evaluates LLMs on multi-task cybersecurity NLP problems such as extracting threat actors or summarizing security reports. While valuable, these benchmarks do not address software-level vulnerability assessment or the detailed technical attributes associated with CVEs.

Our dataset is designed to fill this gap. Beyond extracting and collecting the data, we 1) curate the raw data into detailed portfolios of each CVE across all attributes, and 2) use the portfolios to generate ready-to-use QA pairs that can directly support downstream LLM applications, such as building assistants for security analysts (e.g., an LLM agent that explains root causes or summarizes exploitability), training LLM-based vulnerability triage tools (e.g., models that classify severity, detect duplicates, or map issues to affected components), and powering automated mitigation systems (e.g., agents that propose patch strategies or compare alternative fixes).

## B    Structure of Portfolio

(i) *Data Parsing:* Structured sources such as NVD and Ubuntu Security Notices present data in formats like JSON or are already presented with labels on the website. To capture patch-level data,

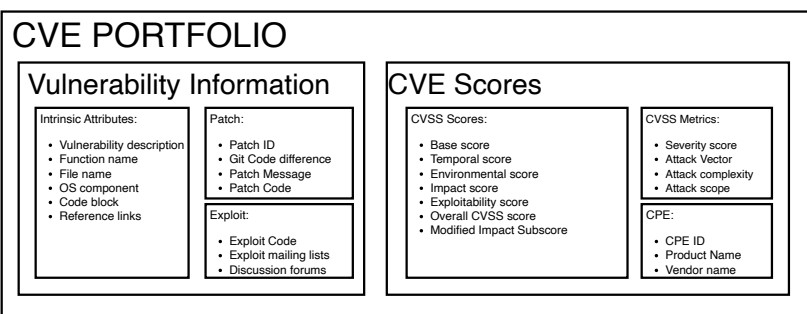

Figure 4: CVE portfolio

we analyzed Git commit histories from `git.kernel.org` and `GitHub.com`. Our parser collected commit messages, code diffs, and identified vulnerable files by isolating modified segments. To obtain pre-patch snapshots, we checked out commits prior to fixes; these serve as inputs for tasks like vulnerability localization and automated patch suggestion with LLMs. From exploit repositories such as Exploit-DB and PacketStormSecurity, we extracted both exploit descriptions and payload code, applying filters to clean metadata and distinguish proof-of-concept (PoC) from weaponized exploits. Unstructured sources—including mailing lists (`Seclists.org`, `lkml.org`), forums (e.g., OpenWall), and exploit archives (`PacketStormSecurity`)—required customized HTML parsers. We downloaded raw pages with Selenium and extracted content blocks with BeautifulSoup. Section 4 presents extraction details of each website.

(ii) *CVE Attribution:* CVE IDs are the canonical keys for aggregating vulnerability records across sources, yet many entries lack explicit CVE fields. In ExploitDB, 41% of items have no CVE tag, and Seclists similarly often includes CVE references only within message text. We therefore recover CVE mentions using lightweight regular expressions over titles, bodies, and comments. Because neither ExploitDB nor Seclists is Linux-specific, we restrict our corpus to Linux-related content using the keyword list in Appendix D. For repositories such as `git.kernel.org` and `GitHub.com`, where CVE attribution is required, we rely on NVD reference links for each CVE as the starting point.

(iii) *Data Aggregation:* In the final step, we aggregate all parsed attributes into unified CVE "portfolios." Using the NVD CVE list as the reference set, we align entries from other sources and retain only CVEs with sufficient context for downstream QA tasks—prioritizing those enriched with exploit details, developer discussions, patch data, and corroborating evidence. This filtering yields 6,342 CVEs, each capable of supporting questions about exploit mechanisms, root causes, and patch details. Furthermore, each portfolio is divided into two major groups: *Vulnerability Information* and *CVE scores.* Each group is further divided into sub-groups that capture fine-grained properties of a CVE. The Vulnerability Information provides the details such as description, patch details, exploit details and so on. On the other hand, the CVE Scores contain all the metrics and related scores such as CVSS, and CPE.

After completing the above steps, we have produced the portfolio and the structure is present in Figure 4.

## C  DATASET DETAILS

We begin with raw vulnerability data collected from the internet and align all artifacts to individual CVEs. To construct a comprehensive Vulnerability Knowledge Base (portfolio), we selected 13 high-value sources based on coverage, artifact richness, and relevance. These include structured repositories (e.g., NVD, Ubuntu, git.kernel.org), unstructured discussion forums (e.g., Seclists, OpenWall, lkml.org), and exploit databases (e.g., Exploit-DB, PacketStormSecurity). We first collected CVEs from NVD, identifying 9,379 Linux entries. Of these, 7,856 include reference links, which provide richer contextual resources for analyzing the corresponding vulnerabilities. Among these, links to official Linux Git repositories were particularly valuable, as they directly contain commits and patches. Using regular expressions, we extracted commit hashes from these

URLs, enabling retrieval of both diffs and commit messages. In total, we collected 24,233 commit messages across 6,325 CVEs, missing hash information for only 107 cases. Next, we incorporated vendor advisories. From `Ubuntu`, we gathered 8,106 CVE pages spanning 2002–2025, and from `RedHat`, we downloaded 6,958 CVE pages. These advisories enrich the dataset with distribution specific scoring, patch status, and impact assessments, complementing the NVD baseline. To capture exploitability information, typically absent from NVD and vendor advisories, we integrated `Exploit-DB`, recovering exploit code for 2,183 CVEs, and `PacketStormSecurity`, which provided artifacts for 63 CVEs. While smaller in volume, these sources capture critical real-world attack vectors.

In addition to the structured websites, there is rich information present in unstructured data sources on the internet, such as mailing lists and developer discussions. We extracted 1,047 threads (2,789 messages) from Seclists, 573 entries from OpenWall, and 154 entries from lkml.org, yielding 3,516 developer discussion entries labeled as *"Developer Discussions"*. These discussions provide insights into patch rationales, root causes, and debates about severity and impact. Exploit-focused repositories such as `Exploit-DB` and `PacketStormSecurity` provide concrete exploit artifacts for 2,183 and 63 CVEs, respectively, which are crucial for understanding real-world attack vectors. For both Exploit-DB and Seclists, we used regular expressions to extract CVE identifiers when they were not explicitly provided.

# D    LINUX KEYWORD

- Linux kernel vulnerability
- Linux kernel CVE
- Linux kernel security issue
- Linux kernel exploit
- Linux kernel patch
- Linux kernel privilege escalation
- Linux kernel remote code execution (RCE)
- Linux kernel syscall vulnerability
- Linux kernel buffer overflow
- Linux kernel memory corruption
- Linux kernel heap overflow
- Linux kernel race condition
- Linux kernel null pointer dereference
- Linux kernel use-after-free
- Linux kernel double free
- Linux kernel information leak
- Linux kernel integer overflow
- Linux kernel arbitrary code execution
- Linux kernel denial of service (DoS)
- Linux kernel local privilege escalation
- `site:seclists.org` Linux kernel CVE / vulnerability / exploit / patch
- `site:seclists.org/oss-sec` Linux kernel CVE
- `site:seclists.org/bugtraq` Linux kernel CVE
- `site:seclists.org/full-disclosure` Linux kernel CVE
- Linux kernel CVE RedHat / Debian / Ubuntu / SUSE
- Linux kernel CVE exploit-db
- Linux kernel CVE MITRE

# E    LINUX GIT REPOSITORIES

- git.kernel.org
- github.com/torvalds/linux
- cgit.kernel.org
- lore.kernel.org
- patchwork.kernel.org

# F    Implementation Details of the RAG system:

## F.1    RAG information:

The major components in the RAG system are as follows:

- Dataset Chunking: converting large text corpus into different segments.

- Embeddings Generation: convert the textual data into embeddings for similarity and faster retrieval.

- Similarity Search: Identifying the similar chunks that are related to a specific query.

- Large Language Model: This uses the retrieved chunks to generate a response for a given query. We have used Langchain (Chase & contributors, 2022) python library for implementing the architecture of our RAG system. In specific, we used LangChain's RecursiveCharectorTextSplitter to break the large amount of data corpus per CVE into smaller chunks for easier processing. These chunks are then converted into OpenAI embeddings for better representation in Vector space and are stored using Facebook's Similarity search for vectors (FAISS) technique (Johnson et al., 2019). To select the relevant chunks from this vector store, we have used Maximal Marginal Relevance (Carbonell & Goldstein, 1998) to select chunks that are relevant to the query and also are different from each other. This helps us in minimizing the selection of any duplicate data from the text corpus. The LLM used in RAG pipeline is GPT-4o for its superior reasoning capabilities at the time of generating the datasets. To further improve determinism, the temperature of the LLM is fixed at 0, eliminating unintended randomness in the generated responses.

In addition to GPT-4o, other LLMs such as Gemini 1.5 Pro and Llama 3 were also evaluated. GPT-4o achieved the strongest performance, so we selected it for our experiments. Below is an example of an exploit explanation for the CVE "CVE-2002-2438," illustrating that GPT-4o provides a clearer and more detailed account of the exploit across multiple dimensions:

**GPT-4o**

```
TCP firewalls could be circumvented by sending SYN packets with other flags,
such as the RST flag, set. These packets were not correctly discarded by the
Linux TCP stack after firewalling, allowing the attacker to bypass firewall
rules. Additionally, YN+FIN attacks were reported to bring Linux machines to
their limits, indicating a potential denial-of-service condition. The
protection added in Linux 2.4.20 involved checking for the RST flag when a
SYN packet is received, which was the main attack scenario.
```

**Llama 3**

```
TCP firewalls could be circumvented by sending a SYN packet with other flags
(like e.g. FIN flag) set, which was not correctly discarded by the Linux TCP
stack after irewalling.
```

**(Gemini 1.5 Pro)**

```
Not found.
```

Another important reason for selecting GPT-4o is its strong reasoning capabilities. Other models with comparable or superior reasoning abilities could also be used in principle.

## F.2    System Prompt used for LLM in RAG pipeline

The system prompt that is used in the LLM component of our RAG system is presented in Figure 5.

## F.3    Sample RAG user prompt

The user prompt for "Vulnerability Description" is presented in Figure 6. Similar prompts are created for all 23 attributes in the RAG system to extract the QA pairs.

---

**System Prompt for RAG pipeline**

You are an information extraction and comprehension agent working for a security analyst. Your role is to analyze CVE (Common Vulnerabilities and Exposures) and CVSS details from a given text corpus either extract or comprehend specific attribute details with precision and completeness.

**Instructions**:
Ensure the output is strictly derived from the given content.
- Do not include any introductory phrases, headings, or comments in your response.

**Final instructions**:
Before generating the answer, carefully check that the detail exists in the context and is correctly identified. Always verify that no assumptions are being made. Use precise, formal language suitable for security documentation. Do not speculate, infer, or assume technical details.

Figure 5: System Prompt used in RAG

---

**Sample RAG query (Vulnerability description)**

Identify any vulnerability that is explicitly described in the context. If present, write a clear, concise `vulnerability_description` in paragraph form. The paragraph may include any of the following details, but only if they are stated in the context:

- What the vulnerability is (e.g., a use-after-free, buffer overflow, race condition)

- The root cause or failure that leads to it

- The conditions or environment required to trigger it (e.g., namespace setup, user privileges)

- The affected components, files, functions, structures, or flags

- The potential impact or consequence of the vulnerability (e.g., privilege escalation, memory corruption)

Your description should focus solely on the weakness or flaw itself — not how it is exploited or what the attacker might do. Do not include proof-of-concept (PoC), exploitation techniques, or patch information.
If no vulnerability is described in the context, respond with: **Not found**

Example of a valid Response:
A logic flaw in the memory subsystem allows a non-stateful expression to be added to a set and subsequently destroyed without proper cleanup. This causes a dangling pointer to remain in the bindings list, leading to a use-after-free condition. The issue occurs when `nft_set_elem_expr_alloc()` initializes an expression that fails the NFT_EXPR_STATEFUL check.
Do not infer, assume, complete, or invent any technical details that are not clearly and explicitly stated in the context.

Figure 6: User prompt example in RAG

Table 7: Questions used to evaluate SOTA models

| Key | Question |
|---|---|
| Vulnerability Identification | Is this program Vulnerable? |
| | Is there a CVE given to this vulnerability? |
| | What is the CVE id? |
| Vulnerable Function Name | What is the name of the vulnerable function affected by the CVE? |
| Vulnerable File Name | What is the name of the file of the OS component that contains the vulnerable function? |
| Vulnerable OS Component | Which kernel component or sub-system is affected with thee vulnerability? |
| Weakness Type | What is the type of the vulnerability? |
| Root Cause | Explain the root cause of the vulnerability? |
| Secure Coding Violations | List and explain any violations of secure coding principles. |
| Mitigation | What are the steps in mitigating this vulnerability? |
| CIA Impact | What is the impact of this vulnerability in terms of Confidentiality, Integrity and Availability? |
| Exploit Explanation | Explain the exploitation technique in detail? |
| Exploit Code | What is the exploit or PoC code that could be used to exploit this vulnerability? |
| Abusable Interfaces | What are the kernel components, device drivers, or interfaces that are involved in triggering or exploiting this vulnerability? |
| Remote Exploitability | Is this vulnerability remotely exploitable? |
| Expliot Steps | List and detail the steps to exploit this vulnerability? |
| Exploit Privileges | What are the privileges that are gained after exploitation? |
| Privileges Required | What are the privileges that are required to exploit this vulnerability? |
| Crash Dump | What is the crash dump or stack trace that shows this vulnerability? |
| Exploited Versions | List the versions of the Linux kernel or distro that are affected by this CVE. |
| Patch Explanation | What does the patch do to fix the issue? |
| Patch Date | When was the patch released? |
| Patched Versions | Which Linux kernel or Linux distribution versions are patched against this vulnerability? |
| Code Difference | What are the git changes that are pushed to patch this vulnerability? |
| Patch Code | What is the patch code? |
| Base Severity | What is the base CVSS severity score of this vulnerability? |
| Base Score | What is the base CVSS score of this vulnerability? |
| Impact Score | What is the impact score for this vulnerability? |
| Exploitability Score | What is the exploitability score? |
| Availability Impact | What is the availability impact? |
| Score Explaination | Explain and comprehend in detail about different scores that are given to this vulnerability so that I can understand the reasoning behind these scores. |

## G Vulnerability Attributes and QA pairs

### G.1 Questions in QA pairs per attribute

The questions in the final QA pairs that represent each attributes are presented in Table 7.

### G.2 List of Vulnerability Attributes

The vulnerability attributes grouped according to the categories are presented in Table 8.

## H Details about Conformal Prediction

The core idea of Conformal Prediction (CP) is to calibrate a nonconformity threshold using a labeled calibration set and then apply that threshold to bound predictive uncertainty on unseen samples. CP has been widely used in classification, regression, and structured prediction tasks for producing statistically valid confidence sets (Angelopoulos & Bates, 2021). We follow the standard split CP approach described in Shafer & Vovk (2008) and Angelopoulos & Bates (2021), which utilizes a

Table 8: Vulnerability Attributes
📄 Descriptions ✇ Deterministic ⟨/⟩ Artifacts

| Category | Attribute | Sub-Attribute |
|---|---|---|
| Primary Attributes | Base Description | ✇ CWE Weakness Type |
| | | 📄 Vulnerability Description |
| | Vulnerability Location | ✇ Vulnerable Function Name, Vulnerable File Name, Vulnerable OS Component |
| | | ⟨/⟩ Vulnerable Code Block |
| | Vulnerability Explanation | 📄 Root Cause, Secure Coding Violation, Mitigation |
| | | ✇ CIA Impact |
| Impact Scores | CVSS Scores | ✇ CVSS Base Score, Impact Subscore, Exploitability Subscore, Overall CVSS Score |
| | | 📄 Scores Description |
| | CPE | ✇ CPE ID, Vendor, Version, Product |
| Exploit | Requirements | ✇ Privileges Required, Affected Versions, Remote Exploitability |
| | Components | ✇ Exploited Interfaces, Abusable Interfaces |
| | Details | 📄 Exploit Description, Exploit Steps |
| | | ⟨/⟩ Exploit Code, Crash Dump |
| | | ✇ Privileges Acquired |
| Patch | Details | ✇ Patch Release Date, Patched Versions |
| | Information | 📄 Patch Description |
| | | ⟨/⟩ Code Difference, Patch Code |

held-out **calibration set** that is assumed to be exchangeable with the test set to estimate non-conformity thresholds for bounding prediction uncertainty.

Formally, let $\mathcal{D}_{\text{cal}} = \{(x_i, s_i^{\text{LLM}}, s_i^{\text{human}})\}_{i=1}^{K}$ be the calibration set of $K$ samples, where $x_i$ is a QA pair, $s_i^{\text{LLM}} \in [1, 5]$ is the evaluation score assigned by the LLM-as-a-Judge, and $s_i^{\text{human}} \in [1, 5]$ is the corresponding human-annotated score. We define the *non-conformity* score for each calibration sample as the absolute difference between LLM and human scores:

$$\delta_i = \left| s_i^{LLM} - s_i^{human} \right| \tag{1}$$

To construct a conformal bound on future deviations, we sort the nonconformity scores in ascending order as $\delta_{(1)} \leq \delta_{(2)} \leq \cdots \leq \delta_{(K)}$, where $(1), \cdots, (K)$ denote the rank indices. Then, given a user-specified significance level $\alpha$, we define the conformal threshold as:

$$\tau_\alpha = \delta_{\lceil (1-\alpha)(K+1) \rceil} \tag{2}$$

where $\lceil (1 - \alpha)(K + 1) \rceil$ indicates the quantile index selected from $(1), \cdots, (K)$ corresponding to the desired coverage level $1 - \alpha$. This threshold represents the maximum deviation between LLM-judged and human-assigned scores that is tolerated to achieve coverage on future test samples. Specifically, for any test sample $x_j^{test}$ with LLM-assigned score $s_j^{LLM}$, we construct a conformal confidence interval given by:

$$\mathcal{I}_j = [s_j^{LLM} - \tau_\alpha, s_j^{LLM} + \tau_\alpha] \cap [1, 5] \tag{3}$$

The interval $\mathcal{I}_j$ is guaranteed to contain the human score with probability at least $1 - \alpha$. In our experiments, we calculate and average the LLM-assigned scores and CP intervals for RAG-generated QA pairs belonging to each attribute, demonstrating that most attributes not only receive high evaluation scores on the test dataset but also exhibit tight confidence bounds and strong alignment with human annotations.

## I   HUMAN EVALUATION DETAILS

The year distribution of 100 CVEs chosen for human evaluation are given in Table 9.

Table 9: Year distribution of the 100 CVEs used in the analysis.

| Year | Count | Year | Count | Year | Count | Year | Count |
|------|-------|------|-------|------|-------|------|-------|
| 2007 | 2 | 2009 | 1 | 2010 | 6 | 2011 | 11 |
| 2013 | 16 | 2014 | 5 | 2015 | 1 | 2016 | 6 |
| 2017 | 4 | 2018 | 5 | 2019 | 9 | 2020 | 12 |
| 2021 | 10 | 2022 | 9 | 2023 | 2 | 2024 | 1 |

Table 10: OS component distribution for the 100 CVEs used in human evaluation.

| OS Component | Count |
|--------------|-------|
| Networking | 30 |
| Filesystems & Storage | 18 |
| Memory Management | 5 |
| Core Kernel | 6 |
| Virtualization (KVM / Xen / vhost) | 8 |
| IPC & Local Communication | 2 |
| Security & Namespaces | 6 |
| Display & Console (TTY / FBdev) | 4 |
| Device Drivers (SPI / USB / misc) | 3 |
| Input Devices | 1 |
| Sound Subsystem | 2 |
| User-space Tools / Applications | 2 |
| Unknown / Not Specified | 13 |

The distribution of Linux subsystems that are spread across the 100 CVEs are presented in Table 10.

A simple intuitive user interface is developed and employed for this operation where the information of the CVE across all the sources is presented on the left and the questions, claims and the buttons are displayed on the right of the screen. The buttons are made bigger to reduce human error. A sample screenshot is provided in Figure 7.

In addition to the two human subjects, another author monitored this experiment as a tie-breaker. If both the human subjects does not agree on a score or to account for any human errors, the tie breaker regularly goes through the answers and resolves all such conflicts. We have got 267 conflicts across 89 CVEs with a conflict percentage of 11.6%. The ground truth labeling involved validating the RAG answers for all the attributes of a CVE with each CVE taking around 30 minutes. This is done for all the chosen 100 CVEs and took 50 hours in total for each human subject. In addition to that the conflict resolution is done in parallel with the ground truth labeling. A total of 10 hours is spent in resolving any conflicts in the ground truth labeling.

To make the human evaluation simpler, we created claims of one sentence for each RAG response. This made it easy for the humans to mark 'yes' or 'no' answers for each claim. A sample of such claims are presented in Figure 8. These claims follow same structure to the ones generated for LLM-as-a-Judge.

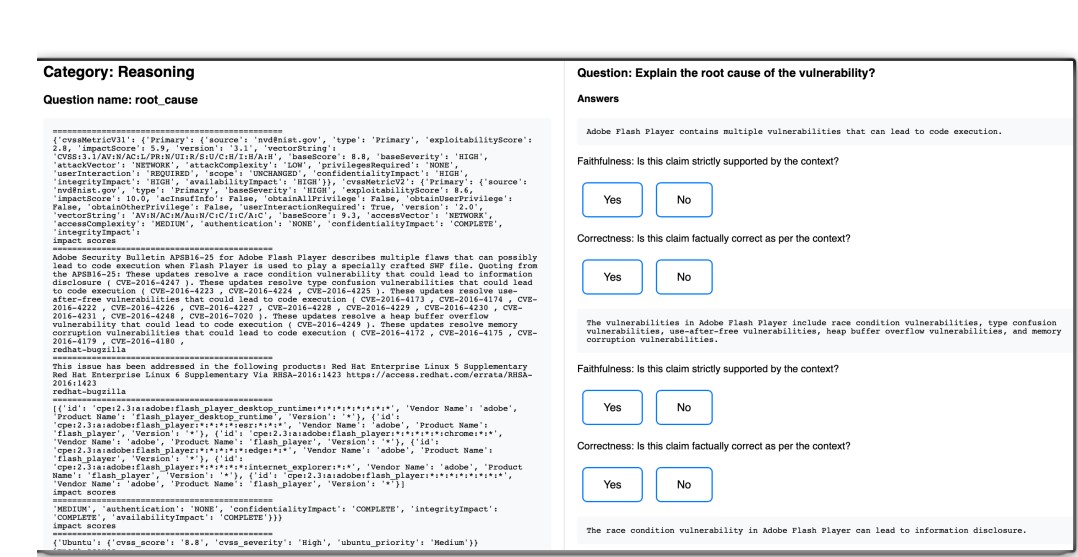

Figure 7: Ground truth labeling UI

### Claims generated from RAG descriptive responses

**Vulnerability Description**

- A vulnerability exists in the IPv6 implementation of the Linux kernel where it does not properly validate socket options in certain situations.
- This flaw allows a local attacker to cause a denial of service or potentially execute arbitrary code.
- The root cause of the vulnerability is related to the improper access and setting of 'raw_sk(sk)→ ipmr_table' before ensuring that the socket is a raw socket and the protocol is IGMP.
- This issue affects Linux distributions using 4.9.x longterm kernels before version 4.9.187.

Figure 8: Example of model-generated claims used in the evaluation

## J DETAILS OF WEIGHTED PRECISION, RECALL AND F1-SCORE

The formulas for calculating *weighted* versions of Precision, Recall, and F1 are given as follows:

$$\text{w-Precision} = \sum_c \frac{n_c}{N} \cdot \text{Precision}_c, \qquad \text{where Precision}_c = \frac{\text{TP}_c}{\text{TP}_c + \text{FP}_c}$$

$$\text{w-Recall} = \sum_c \frac{n_c}{N} \cdot \text{Recall}_c, \qquad \text{where Recall}_c = \frac{\text{TP}_c}{\text{TP}_c + \text{FN}_c}$$

$$\text{w-F1} = \sum_c \frac{n_c}{N} \cdot \text{F1}_c, \qquad \text{where F1}_c = \frac{2 \cdot \text{Precision}_c \cdot \text{Recall}_c}{\text{Precision}_c + \text{Recall}_c}$$

where $n_c$ is the number of samples in class $c$, and $N$ is the total number of samples. These metrics ensure that performance is not biased toward the majority *vulnerable* class and reflect the contribution of each class proportionally to its sample frequency.

## K PERFORMANCE ANALYSIS OF SOTA LLMs WITH SCORES BELOW 3

We selected the attribute *Mitigation* and evaluated LLM-generated outputs for 12 CVEs. Three major limitations identified across CVEs are listed below:

(i) *Overly generic mitigation strategies.* Across multiple CVEs, LLMs defaulted to generic upgrade recommendations, often suggesting a non-vulnerable version rather than providing attribute-specific mitigation guidance.

(ii) *Missing mitigation rationale.* In several cases, the models failed to supply an explanation or actionable remediation steps, instead offering high-level version upgrades even when more direct mitigation options were known to exist.

(iii) *Incorrect details and hallucinations.* The models frequently produced inaccurate Linux version information, distro-specific instructions, and unfounded mitigation details; hallucinations appeared in 4 of the 12 CVEs assessed, with some CVEs receiving inconsistent or conflicting upgrade recommendations across models.

The CVE IDs that have the least mitigation scores and are used for this experiment are as follows:
- CVE-2013-6431    - CVE-2014-7283
- CVE-2019-15031    - CVE-2019-3701
- CVE-2020-12659    - CVE-2020-12769
- CVE-2020-25211    - CVE-2020-27152
- CVE-2021-45868    - CVE-2022-28356
- CVE-2023-40791    - CVE-2023-4921

## L LLM-JUDGE PROMPT FOR RAG-GENERATED OUTPUTS

The prompt for LLM-as-a-Judge assessment of Faithfulness, Correctness, and Completeness regarding RAG-generated CVE attributes is demonstrated in Figure 9.

## M LLM PERFORMANCE EVALUATION PROMPTS

Here, we provide the unified prompt for each SOTA LLM evaluation step: (1) Vulnerability Detection – Figure 10, (2) CVE Identification – Figure 11, and (3) CVE Attribute Question Answering – Figure 12. Additionally, the prompt given in Figure 13 is used for generating web-search queries, which retrieve online CVE information as input for steps 2 and 3 evaluation.

## N SOTA EVALUATION WITH BALANCED DATASETS

To study the impact of class imbalance, we constructed a balanced test set by retaining only CVEs that have both vulnerable and benign code (83 vulnerable and 83 non-vulnerable samples) and

---

### Evaluation Prompt for RAG-generated Attributes

You are an expert evaluator of responses written by a Retrieval-Augmented Generation (RAG) system for questions related to software security and vulnerabilities. You will be given the RAG-generated response, a list of extracted factual claims from the response, the original user question, and the retrieved context used to generate the response. Your task is to rate the RAG response and the extracted claims on three metrics:

– **Faithfulness** - Whether each factual claim is supported by the retrieved context.

– **Correctness** - Whether each factual claim is factually and logically accurate with respect to both the question and the context.

– **Completeness** - Whether the response sufficiently addresses the user question based only on the retrieved context.

**Instructions**:

*Faithfulness*:

– Attribute-Specific scoring rules:
{attribute_rules}

– Default scoring rules:
  ○ Assign *1.0* if the claim is *explicitly stated* or *reasonably supported* in the context.
  ○ Assign *0.5* if the claim is plausible but not fully verifiable or weakly supported by the context.
  ○ Assign *0.0* if the claim is *not supported* or *not verifiable* from the context.

– For each factual claim:
  1. First, check whether *Attribute-Specific rules* apply. If yes, apply them directly and ignore all other rules.
  2. If no Attribute-Specific rule applies, apply the the *Default rules.*
  3. Write a short justification for your faithfulness rating.

*Correctness*:

...

*Completeness*:

...

**Output Format**
Respond with a JSON dictionary object containing three fields:
{json_template}

**Inputs**
*Retrieved context:*
{context}

*Question:*
{question}

*Response to evaluate:*
{response}

*Factual claims to evaluate:*
{claims}

Figure 9: Evaluation Prompt for RAG-generated Attributes

discarding CVEs without a corresponding patch. The results on this balanced dataset are reported in Table 11, where all LLMs still exhibit poor detection performance. Since our main conclusions are driven by the models' inability to reliably identify the vulnerable class, making the dataset more balanced does not change our findings about the limitations of current LLMs for vulnerability detection.

---

**Step 1 –Vulnerability Detection Prompt**

**Role and Objective:**
You are an expert software security analyst. You will be given a list of dictionaries, where each key is a file name and each value is a code block within that file.
Analyze only the code blocks and decide whether each code block contains a security vulnerability.

**Definitions:**

- "vulnerable": The code exhibits a concrete, actionable security flaw (e.g., memory safety bug, injection, auth/crypto misuse) that is inferable from the snippet itself.
- "not_vulnerable": No clear vulnerability is present based solely on the snippet.

**Instructions:**
For each file name:

- Return one of: 1 (vulnerable), or 0 (not_vulnerable).
- Use the visible code as the primary evidence. Do NOT invent vulnerabilities unrelated to the snippet.
- If you mark 1 (vulnerable), you must internally identify at least one concrete flaw in the snippet (but do not explain it in the output).

**Output Format (strict):**
Respond with a JSON object where each file name from the input dictionary is a key and the value is its classification:
{
"FileA.c": 1,
"FileB.h": 0,
...
}

Do NOT include any explanations, Markdown, or code fences. Output raw JSON only.

**Code Blocks:**
{code_blocks}

Figure 10: Step 1 –Vulnerability Detection Prompt

## Step 2 – CVE Identification

**Role and Objective:**
You are an expert software security analyst.
You will be given:

1. A list of file names belonging to an unknown CVE ID.

2. A corresponding list of dictionaries where each key is a file name and each value is a vulnerable code block from that file.

3. A set of pre-fetched search results relevant to the vulnerabilities in the code blocks.

These code blocks have ALREADY been identified as containing security vulnerabilities.
Your task is to analyze the vulnerable file names, code blocks, and optionally the search results, then retrieve exactly 5 CVE IDs that could match the vulnerabilities evidenced by the code.

**Instructions:**

– Use the file names and visible code blocks (including in-code strings/comments) as the primary evidence; also use the search results to corroborate, refine, or downgrade candidates.

– If the search results is empty or unhelpful, rely on the file names, code blocks, and your broader knowledge of publicly documented CVEs.

– Treat the file names as meaningful context, since they may correspond to OS components, kernel modules, or libraries related to CVE IDs.

– Strong signals include: function names, file names, protocol/feature identifiers, exploit strings, or in-code CVE references.

– Always produce exactly 5 CVE candidates for the given input (not per file):

   ○ 0.85–1.00 = strong match (near-unique trigger + CWE/vulnerability type + component corroborated by search results)

   ○ 0.60–0.84 = moderate match (strong overlap, partial corroboration)

   ○ 0.40–0.59 = weak match (generic overlap or minimal corroboration)

   ○ below 0.40 = very weak match (include if needed to fill 5 slots)

– Rank the candidates from highest to lowest confidence.

– Exclude RESERVED/REJECT CVE IDs and duplicates/aliases.

– Do NOT fabricate CVE IDs, only output CVE IDs that are real.

**Output Format (strict):**
Respond with a JSON array containing exactly 5 objects:

```
[
{
"CVE_ID": "CVE-YYYY-NNNN",
"confidence": < float >,
"match_reason": "short reason"
},
...
]
```

Do NOT include any explanations, Markdown, or code fences. Output raw JSON only.

**File Names:**
{file_names}

**Vulnerable Code Blocks:**
{code_blocks}

**Search Results:**
{search_context}

Figure 11: Step 2 – CVE Identification

## Step 3 – CVE Attribute Question-Answering

**Role and Objective:**
You are an expert software security analyst.
You will be given:

1. A CVE ID.

2. A corresponding list of file names belonging to that CVE ID.

3. A corresponding list of dictionaries where each key is a file name and each value is a vulnerable code block from that file.

4. A set of pre-fetched search results relevant to the vulnerabilities in the code blocks.

These code blocks have ALREADY been identified as containing security vulnerabilities.
Your task is to answer the user question about the CVE with precision and completeness.

**Instructions:**

– Use all available information: the CVE ID, file names, visible code blocks, and the search results as evidence.

– You may also incorporate your broader knowledge of publicly documented CVEs, particularly the given CVE ID, from sources such as vulnerability databases, advisories, and security literature.

– If the question explicitly asks for code (e.g., "vulnerable code", "exploit code", "patch code", "code diff", "crash dump", etc.):

  ○ Always include the actual code snippet(s) from the provided input if available.
  ○ If the relevant code is not in the provided input, you may use your broader knowledge of the CVE ID to recall the correct code from well-documented advisories or patches.
  ○ If you cannot reliably recall the exact code, respond with "Not found".

– For all *non-code answers*, keep the response *under 150 words*.

– Provide precise, factual, and complete answers in formal language suitable for security documentation.

– If the answer cannot be determined from the CVE ID, file names, visible code blocks, the search results, or your broader knowledge of publicly documented CVEs, reply with "Not found".

**CVE ID:**
{cve_id}

**File Names:**
{file_names}

**Vulnerable Code Blocks:**
{code_blocks}

**Search Results:**
{search_context}

**Question:**
{question}

Figure 12: Step 3 – CVE Attribute Question-Answering

---

**Web Search Prompt**

**Role and Objective:**
You are an expert software security analyst.
You will be given:

1. A list of file names belonging to an unknown CVE ID.

2. A corresponding list of dictionaries where each key is a file name and each value is a vulnerable code block from that file.

These code blocks have ALREADY been identified as containing security vulnerabilities.
Your task is to propose focused web search queries to find the most likely matching CVE IDs.

**Instructions:**

- Return up to 5 UNIQUE queries; each query should be concise (under 15 words).

- Build queries by combining exact identifiers from the snippets (component/module names, file paths, function names, error strings) with the word CVE.

- Prefer site filters to authoritative sources: NVD, MITRE, vendor advisories, kernel.org/git commits, LKML, Debian/Ubuntu/Red Hat advisories.

- Extract only minimal facts (CVE ID, component/module, CWE/type, patch/commit IDs, affected versions).

- Do NOT include links or citations in the output.

**Output Format (strict):**
Respond with a JSON list of strings:
[
"Query 1",
"Query 2",
...
]

Do NOT include any explanations, Markdown, or code fences. Output raw JSON only.

**File Names:**
{file_names}

**Vulnerable Code Blocks:**
{code_blocks}

---

Figure 13: Web Search Prompt

Table 11: Vulnerability detection results on a balanced subset (83 vulnerable vs. 83 non-vulnerable samples).

| Model | Precision | Recall | F1 |
|---|---|---|---|
| GPT-4o | 48.79 | 48.80 | 48.78 |
| Claude 4 | 48.35 | 48.80 | 45.11 |
| Gemini 2.5 | 28.72 | 45.18 | 32.04 |
| Qwen 3 | 51.82 | 51.81 | 51.70 |
| Llama 3.3 | 58.90 | 55.43 | 38.78 |

Table 12: CVE Attribute Question-Answering Evaluation Using Optimized Prompts for Claude 4 and Llama 3.3. MA - Model-Agnostic, O - Optimized

| Attribute | Claude 4-MA | Claude 4-O | Llama 3.3-MA | Llama 3.3-O |
|---|---|---|---|---|
| Function Name | 3.16 | 2.93 | 3.74 | **3.78** |
| File Name | 3.53 | 3.12 | 3.78 | **3.79** |
| Vulnerable OS Components | 4.50 | **4.63** | 4.07 | **4.38** |
| Weakness Type | 3.30 | **3.72** | 3.31 | **3.53** |
| Root Cause | 4.65 | 4.50 | 4.42 | 4.42 |
| Secure Coding Violations | 4.06 | 3.96 | 3.10 | 2.69 |
| Mitigation | 2.05 | **3.04** | 1.91 | **2.28** |
| CIA Impact | 4.32 | 3.56 | 4.12 | 2.89 |
| Exploit Explanation | 3.00 | **3.33** | 2.77 | **2.84** |
| Exploit Code | 3.02 | **3.67** | 3.62 | **3.80** |
| Exploited Versions | 2.59 | 3.03 | 2.57 | **2.76** |
| Abusable Interfaces | 3.06 | **3.88** | 3.19 | **3.40** |
| Exploit Steps | 3.51 | 3.21 | 3.28 | 3.24 |
| Privileges Required | 4.33 | 4.00 | 3.49 | 3.32 |
| Exploit Privileges | 4.34 | **4.41** | 2.65 | **4.48** |
| Remote Exploitability | 4.56 | 3.04 | 4.48 | 3.88 |
| Crash Dump | 1.74 | **3.90** | 3.88 | 3.88 |
| Patch Explanation | 3.89 | 2.25 | 3.38 | 2.28 |
| Patch Date | 1.52 | **2.50** | 1.88 | **2.30** |
| Patched Versions | 1.79 | **2.14** | 1.64 | **2.26** |
| Code Difference | 1.83 | **2.73** | 2.50 | **2.76** |
| Patch Code | 2.54 | 1.41 | 1.28 | **1.42** |
| Score Explanations | 3.79 | **3.94** | 3.25 | 3.18 |

## O    EVALUATION OF SOTA LLMs WITH OPTIMIZED PROMPTS

The results of the evaluation of SOTA LLMs (Claude and LLmma) using optimized prompts are presented in Table 12.

## P    EVALUATION OF SOTA LLMs WITH IN-CONTEXT LEARNING

The results of the CVE attribute QA evaluation using GPT-4o with 3-shot in-context learning are summarized in Table 13. We can observe that adding only a few in-context examples improves Correctness on most CVE attributes (20/23), with particularly large gains for reasoning-heavy attributes such as Secure Coding Violations, Mitigation, and Patch Code. These results confirm that our dataset is indeed useful for enhancing LLM performance when used as supervision.

## Q    EVALUATION OF SOTA LLMs WITH LESS-DOCUMENTED CVEs

The results of the CVE attribute QA evaluation using GPT-4o with 10 less-documented CVEs are summarized in Table 14.

Table 13: CVE attribute QA results using 3-shot in-context learning with GPT-4o. Here, **boldface** denotes where in-context learning improves the Correctness score compared to the zero-shot setting.

| Attribute | Zero-shot | In-context Learning |
|---|---|---|
| Function Name | 3.67 | **3.70** |
| File Name | 3.85 | **3.77** |
| Vulnerable OS Components | 3.96 | **4.43** |
| Weakness Type | 3.02 | **3.29** |
| Root Cause | 4.22 | **4.35** |
| Secure Coding Violations | 3.57 | **4.00** |
| Mitigation | 2.11 | **2.54** |
| CIA Impact | 3.92 | **3.56** |
| Exploit Explanation | 2.69 | **2.72** |
| Exploit Code | 3.70 | **3.79** |
| Exploited Versions | 2.94 | **3.34** |
| Abusable Interfaces | 3.61 | **3.71** |
| Exploit Steps | 3.40 | **3.41** |
| Privileges Required | 3.77 | 3.41 |
| Exploit Privileges | 3.03 | **4.32** |
| Remote Exploitability | 4.39 | **4.65** |
| Crash Dump | 3.09 | **3.95** |
| Patch Explanation | 3.50 | **3.75** |
| Patch Date | 1.31 | **1.76** |
| Patched Versions | 1.84 | **1.97** |
| Code Difference | 2.53 | **2.68** |
| Patch Code | 1.50 | **2.06** |
| Score Explanations | 2.78 | **2.95** |

Table 14: CVE attribute question-answering evaluation with GPT-4o using 10 less documented CVEs. Here, **boldface** indicates where less documented CVEs have lower Correctness score for a CVE attribute.

| CVE Attribute | 200 Rich-Context CVEs | 10 Less-Documented CVEs |
|---|---|---|
| Function Name | 3.67 | **2.20** |
| File Name | 3.85 | **3.80** |
| Vulnerable OS Components | 3.96 | 4.30 |
| Weakness Type | 3.02 | **2.40** |
| Root Cause | 4.22 | **2.56** |
| Secure Coding Violations | 3.57 | **3.40** |
| Mitigation | 2.11 | 2.40 |
| CIA Impact | 3.92 | **3.10** |
| Exploit Explanation | 2.69 | **2.30** |
| Exploit Code | 3.70 | **3.40** |
| Exploited Versions | 2.94 | **2.50** |
| Abusable Interfaces | 3.61 | **3.20** |
| Exploit Steps | 3.40 | **2.80** |
| Privileges Required | 3.77 | 3.80 |
| Exploit Privileges | 3.03 | 3.90 |
| Remote Exploitability | 4.39 | **4.00** |
| Crash Dump | 3.09 | **2.00** |
| Patch Explanation | 3.50 | **3.44** |
| Patch Date | 1.31 | 1.70 |
| Patched Versions | 1.84 | 2.60 |
| Code Difference | 2.53 | **1.80** |
| Patch Code | 1.50 | **1.40** |
| Score Explanations | 2.78 | 2.90 |

Table 15: CVE identification result using GPT-4o when vulnerability detection and identification evaluation steps are combined.

| CVE Subset | Top1 Accuracy | Top5 Accuracy |
|---|---|---|
| True Positives | 53.77 | 73.58 |
| False Positives | 34.09 | 52.27 |
| Overall | 48.00 | 67.33 |

Table 16: Human-annotated scores averaged over 100 human assessment samples for 23 RAG-generated CVE attributes.

| Attribute | Faithfulness | Correctness | Completeness |
|---|---|---|---|
| Function Name | 4.88 | 4.88 | 4.80 |
| File Name | 4.96 | 4.96 | 4.94 |
| Vulnerable OS Components | 5.00 | 5.00 | 4.99 |
| Weakness Type | 4.88 | 4.88 | 4.87 |
| Root Cause | 5.00 | 5.00 | 5.00 |
| Secure Coding Violations | 4.92 | 4.92 | 4.96 |
| Mitigation | 4.86 | 4.86 | 4.92 |
| CIA Impact | 4.70 | 4.71 | 4.88 |
| Exploit Explanation | 4.96 | 4.96 | 4.95 |
| Exploit Code | 4.40 | 4.36 | 4.40 |
| Exploited Versions | 4.98 | 4.93 | 4.99 |
| Abusable Interfaces | 4.80 | 4.80 | 4.80 |
| Exploit Steps | 5.00 | 5.00 | 4.98 |
| Privileges Required | 4.84 | 4.84 | 4.82 |
| Exploit Privileges | 4.96 | 4.96 | 4.94 |
| Remote Exploitability | 5.00 | 5.00 | 5.00 |
| Crash Dump | 4.84 | 4.84 | 4.81 |
| Patch Explanation | 4.92 | 4.92 | 4.92 |
| Patch Date | 4.96 | 4.96 | 4.96 |
| Patched Versions | 4.96 | 4.96 | 4.95 |
| Code Difference | 4.88 | 4.87 | 4.82 |
| Patch Code | 4.40 | 3.98 | 4.04 |
| Score Explanations | 4.99 | 4.99 | 5.00 |

## R  EVALUATION OF COMBINED VULNERABILITY DETECTION PHASE AND IDENTIFICATION PHASE

The results of combined vulnerability detection and CVE identification phases are presented in Table 15.

## S  AVERAGE HUMAN ANNOTATION SCORES FOR 100 CVEs

The human annotated scores across three metrics averaged for all 100 CVEs are presented in Table 16.

## T  THE USE OF LARGE LANGUAGE MODELS (LLMs)

We use LLMs to polish our writings.

