# OpenReview forum: "DiagVuln: A Holistic Conversational Benchmark for Evaluating LLMs on Vulnerability Assessment"
_ICLR.cc/2026/Conference — ICLR 2026 Conference Desk Rejected Submission_

### Official Review · Reviewer_kj1d · 2025-10-16

**Soundness:** 3
**Presentation:** 3
**Contribution:** 2
**Rating:** 4
**Confidence:** 3

**Summary:**

The paper proposes a new vulnerability dataset including CVEs in the Linux kernel. The raw information was collected from NVD as well as public websites and forums, and RAG was used to create question answer pairs about the vulnerabilities. The quality of the generated QA pairs is validated using human effort. Finally, the paper benchmarks existing SOTA commercial and open-source LLMs on the proposed dataset.

**Strengths:**

- The ability of the RAG method to generate QA pairs is sufficiently validated by the human assessment.
- The collected dataset includes useful information that could boost the performance of LLMs, either by using RAG or fine-tuning.

**Weaknesses:**

- Overall the paper does not well-motivate the need for the proposed dataset. The results section did not show any comparison with previous datasets (e.g., in base LLM performance, or in how training on the dataset improves performance).
- It seems natural to evaluate to which extent augmenting an LLM with the knowledge in the proposed dataset (through RAG or fine-tuning) would improve the performance of LLMs in the task. The authors mention this as a potential future work, but in my opinion this should have been included in the paper.
- The paper only benchmarks LLMs, without mentioning other vulnerability detection/identification tools. Both the detection and identification tasks could be carried out by either static analyzers or by DNN-based methods.
- The related work section did not sufficiently mention existing vulnerability detection and assessment datasets, and how the proposed dataset is different/better. This is currently mentioned in the introduction, but only very briefly
- The separate evaluation of vulnerability detection and identification might be misleading. I think all false positives from the detection step should also be considered in the input to the identification step, which should significantly degrade the identification results.

**Questions:**

- How does your proposed dataset help push the state of the art in vulnerability detection and assessment?
- Did you only focus on the vulnerabilities in the Linux kernel project, or do you include any vulnerabilities that affect Linux systems?
- Related to the above point, why did you only focus on Linux kernel vulnerabilities? How does this affect the applicability of your dataset?
- Why did not you benchmark other vulnerability detection/identification methods (e.g., static analyzers, GNNs, RNNs)?
- Could you elaborate more on the developed RAG system used to create the dataset? What was the knowledge base items? what were the queries used?
- What was the source for benign code in your dataset? How would having a more balanced (benign vs vulnerable ratio) dataset affect the results in Section 5?
- How would unifying the detection and identification pipeline affect the results in Section 5?

---

> ### Author Response · Authors · 2025-11-23
> **Response to Reviewer kj1d (1/6)**
>
> Q1**: **Motivation and Contribution**
> *Overall the paper does not well-motivate the need for the proposed dataset. The results section did not show any comparison with previous datasets (e.g., in base LLM performance, or in how training on the dataset improves performance).*
> *How does your proposed dataset help push the state of the art in vulnerability detection and assessment?*
>
> **A1**: Recent work [1,2,3,4,5]  increasingly explores the use of LLMs for vulnerability detection, triage, and reasoning, reflecting the community’s interest in harnessing LLM capabilities for software security. However, existing studies also highlight substantial limitations: LLMs frequently hallucinate vulnerability facts [6], misinterpret code semantics [7], or fail to follow security-critical constraints [8].
> Despite these findings, there is still no comprehensive, fine-grained benchmark that can help systematically evaluate LLM performance across the full vulnerability-assessment workflow, from detection to attribution, localization, patch explanation, and reasoning. This gap directly motivates our dataset. Prior datasets either (i) focus solely on code-level vulnerability classification [9], (ii) cover only CVE text without associated code [10], or (iii) provide no reasoning-oriented annotations [11]. None of them support multi-step evaluation of how and why an LLM identifies a vulnerability, nor do they allow consistent comparison across models.
>
> *Our contributions:*
>
> Our dataset is designed to fill this gap. Beyond extracting and collecting the data, we 1) curate the raw data into detailed portfolios of each CVE across all attributes, and 2) use the portfolios to generate ready-to-use QA pairs that can directly support downstream LLM applications, such as building assistants for security analysts (e.g., an LLM agent that explains root causes or summarizes exploitability), training LLM-based vulnerability triage tools (e.g., models that classify severity, detect duplicates, or map issues to affected components), and powering automated mitigation systems (e.g., agents that propose patch strategies or compare alternative fixes).
>
> As pointed by *reviewer h3gR*, our dataset makes valuable contributions: "It is a valuable benchmark for evaluating the possible extent of use of LLMs in critical cybersecurity situations and the fine-grained questions provide more insight than previous benchmarks".
>
> [1] Shimmi, Samiha, Ashiqur Rahman, Mohan Gadde, Hamed Okhravi, and Mona Rahimi. “{VulSim}: Leveraging Similarity of {Multi-Dimensional} Neighbor Embeddings for Vulnerability Detection.” 2024, 1777–94.
>
> [2] Shimmi, Samiha, Yash Saini, Mark Schaefer, Hamed Okhravi, and Mona Rahimi. “Software Vulnerability Detection Using LLM: Does Additional Information Help?” 2024 Annual Computer
> Security Applications Conference Workshops (ACSAC Workshops), December 2024, 216–23.
>
> [3] Lekssays, A., Mouhcine, H., Tran, K., Yu, T., & Khalil, I. (2025). {LLMxCPG}:{Context-Aware} vulnerability detection through code property {Graph-Guided} large language models. In 34th USENIX Security Symposium (USENIX Security 25) (pp. 489-507).
>
> [4] Yang, Chenyuan, Zijie Zhao, Zichen Xie, Haoyu Li, and Lingming Zhang. “KNighter: Transforming Static Analysis with LLM-Synthesized Checkers.” Proceedings of the ACM SIGOPS 31st Symposium on Operating Systems Principles, October 13, 2025, 655–69.
>
> [5] Fu, Michael, Chakkrit Kla Tantithamthavorn, Van Nguyen, and Trung Le. “ChatGPT for Vulnerability Detection, Classification, and Repair: How Far Are We?” IEEE Computer Society, December 1, 2023, 632–36.
>
> [6] Spracklen, Joseph, and Murtuza Jadliwala. "Package Hallucinations: How LLMs Can Invent Vulnerabilities."
>
> [7] Nikiema, Serge Lionel, et al. "The Code Barrier: What LLMs Actually Understand?." arXiv preprint arXiv:2504.10557 (2025).
>
> [8] Khoury, Raphaël, et al. "How secure is code generated by chatgpt?." 2023 IEEE international conference on systems, man, and cybernetics (SMC). IEEE, 2023.
>
> [9] Zhou, Y., Liu, S., Siow, J., Du, X. & Liu, Y. (2019). Devign: Effective Vulnerability Identification by Learning Comprehensive Program Semantics via Graph Neural Networks. In Proceedings of the 33rd International Conference on Software Engineering, pp. 1013-1024.
>
> [10] Guru Bhandari, Amara Naseer, and Leon Moonen. 2021. CVEfixes: automated collection of vulnerabilities and their fixes from open-source software. In Proceedings of the 17th International Conference on Predictive Models and Data Analytics in Software Engineering (PROMISE 2021). Association for Computing Machinery, New York, NY, USA, 30–39.
>
> [11] Yangruibo Ding, Yanjun Fu, Omniyyah Ibrahim, Chawin Sitawarin, Xinyun Chen, Basel Alomair, David Wagner, Baishakhi Ray, and Yizheng Chen. 2025. Vulnerability Detection with Code Language Models: How Far Are We? In Proceedings of the IEEE/ACM 47th International Conference on Software Engineering (ICSE'25). IEEE Press, 1729–1741.

---

> ### Author Response · Authors · 2025-11-23
> **Response to Reviewer kj1d (2/6)**
>
> **Q2** **Measure the performance of a fine-tuned model**
> *It seems natural to evaluate to which extent augmenting an LLM with the knowledge in the proposed dataset (through RAG or fine-tuning) would improve the performance of LLMs in the task. The authors mention this as a potential future work, but in my opinion this should have been included in the paper.*
>
> **A2**: We have conducted an in-context learning experiment to examine the effect of augmenting an LLM with the knowledge in our proposed dataset. Specifically, in the CVE attribute question-answering (QA) experiment using GPT-4o, we provide 3 QA pairs as in-context examples for each attribute. The results summarized in Table 1 below  compare the Correctness scores in this 3-shot in-context setting against the original zero-shot setting (Section 5). We observe that adding only a few in-context examples improves Correctness on most CVE attributes (20/23), with particularly large gains for reasoning-heavy attributes such as Secure Coding Violations, Mitigation, and Patch Code. These results confirm that our dataset is indeed useful for enhancing LLM performance when used as supervision. We will report these in-context learning results in the Appendix.
>
> Table 1: CVE attribute QA results using 3-shot in-context learning with GPT-4o. Here, **boldface** denotes where in-context learning improves the Correctness score compared to the zero-shot setting.
>
> | **Attribute**            | Zero-shot | In-context Learning|
> |--------------------------|-----------|----------------------------------|
> | Function Name            | 3.67      | **3.7**                          |
> | File Name                | 3.85      | 3.77                             |
> | Vulnerable OS Components | 3.96      | **4.43**                         |
> | Weakness Type            | 3.02      | **3.29**                         |
> | Root Cause               | 4.22      | **4.35**                         |
> | Secure Coding Violations | 3.57      | **4**                            |
> | Mitigation               | 2.11      | **2.54**                         |
> | CIA Impact               | 3.92      | 3.56                             |
> | Exploit Explanation      | 2.69      | **2.72**                         |
> | Exploit Code             | 3.70      | **3.79**                         |
> | Exploited Versions       | 2.94      | **3.34**                         |
> | Abusable Interfaces      | 3.61      | **3.71**                         |
> | Exploit Steps            | 3.40      | **3.41**                         |
> | Privileges Required      | 3.77      | 3.41                             |
> | Exploit Privileges       | 3.03      | **4.32**                         |
> | Remote Exploitability    | 4.39      | **4.65**                         |
> | Crash Dump               | 3.09      | **3.95**                         |
> | Patch Explanation        | 3.50      | **3.75**                         |
> | Patch Date               | 1.31      | **1.76**                         |
> | Patched Versions         | 1.84      | **1.97**                         |
> | Code Difference          | 2.53      | **2.68**                         |
> | Patch Code               | 1.50      | **2.06**                         |
> | Score Explanations       | 2.78      | **2.95**                         |
>
>
> **Q3: Comparison with Vulnerability detection techniques**
> *The paper only benchmarks LLMs, without mentioning other vulnerability detection/identification tools. Both the detection and identification tasks could be carried out by either static analyzers or by DNN-based methods.*
> *Why did not you benchmark other vulnerability detection/identification methods (e.g., static analyzers, GNNs, RNNs)?*
>
> **A3**: Our goal in this work is not to benchmark against classical vulnerability detection tools, but to evaluate the capability of modern LLMs in end-to-end vulnerability assessment. This includes not only detection, but also reasoning about root causes, understanding exploitability, and generating remediation suggestions, where traditional static analyzers (e.g., CodeQL) and DNN-based models are not designed to provide. For this reason, we compare LLMs against each other to surface differences in their security reasoning ability, rather than against static analyzers whose goals and output formats are fundamentally different.

---

> ### Author Response · Authors · 2025-11-23
> **Response to Reviewer kj1d (3/6)**
>
> **Q4: Related work**
> *The related work section did not sufficiently mention existing vulnerability detection and assessment datasets, and how the proposed dataset is different/better. This is currently mentioned in the introduction, but only very briefly*
>
> **A4**: The Table 1 of the paper details the previous datasets and how our proposed dataset is different from the previous works. We will add detailed comparison with the existing vulnerability detection and assessment datasets as follows:
>
> (1) Vulnerability data sources
> There is no comprehensive, fine-grained benchmark that systematically evaluates LLM performance across the full vulnerability-assessment workflow, from detection to attribution, localization, patch explanation, and reasoning. Prior datasets either (i) focus solely on code-level vulnerability classification [1], (ii) cover only CVE text without associated code [2], or (iii) provide no reasoning-oriented annotations [3]. None of them support multi-step evaluation of how and why an LLM identifies a vulnerability, nor do they allow consistent comparison across models.
>
> (2) Dataset dimensionality:
> Additionally, prior attempts to build cybersecurity datasets frequently suffer from small data volume or rely on multiple-choice question formats, which constrain task diversity and limit their usefulness for instruction tuning. These evaluations also depend on narrow datasets or short LLM memory, restricting scalability and preventing support for multi-turn conversations.
>
> (3) Data quality and correctness:
> While there are other CVE datasets that were presented [2,3,4,5,6,7] most of them neither propose a concrete evaluation procedure nor have a scalable approach for evaluation even when the dataset is labeled using automated techniques. DiverseVul [7] contains direct crawled data from internet and acknowledged that the proposed dataset might have a limitation of label noise. In addition to [7], other datasets [2,4,5,7] were also shown to be containing mislabeled vulnerable samples [3]. PrimeVul [3] offers the most systematic evaluation to date, demonstrating substantial reductions in label noise across previously released datasets, including its own. However, its evaluation process relies heavily on manual inspection and therefore does not scale to larger corpora. The latest work [6] provides an automated and scalable data collection pipeline but did not have a concrete evaluation strategy to assess the quality of the dataset.
>
> In response to these gaps, we introduce a dataset that combines scalability with a clearly articulated evaluation protocol. Our assessment framework employs both human review and LLM-as-a-Judge strategy to systematically measure and validate labeling accuracy. Even though our manual evaluation of 100 CVEs with 3 human subjects took around 60 human hours, this can be used in conjunction with LLM-as-a-Judge for scaling up to further new samples.
>
> (4) Suited for LLM training and evaluation:
> A series of recent evaluations— Purple Llama CyberSecEval [8], CyberSecEval 2 [9], and CyberSecEval 3 [10]—focus on assessing LLM behavior in security-sensitive contexts such as insecure code generation, cyberattack assistance, interpreter abuse, prompt injection, and broader offensive capabilities. CyberBench [11] further evaluates LLMs on multi-task cybersecurity NLP problems such as extracting threat actors or summarizing security reports. While valuable, these benchmarks do not address software-level vulnerability assessment or the detailed technical attributes associated with CVEs.
>
> We compile multi-source details such as technical attributes, discussions, exploit details, and patch rationales into structured QA pairs suitable for fine-tuning and benchmarking LLMs. This creates a more complete foundation for evaluating and training LLMs on end-to-end vulnerability assessment and supports deeper reasoning and practical real-world applicability.

---

> ### Author Response · Authors · 2025-11-23
> **Response to Reviewer kj1d (4/6) - References for above comment**
>
> [1] Zhou, Y., Liu, S., Siow, J., Du, X. & Liu, Y. (2019). Devign: Effective Vulnerability Identification by Learning Comprehensive Program Semantics via Graph Neural Networks. In Proceedings of the 33rd International Conference on Software Engineering, pp. 1013-1024.
>
> [2] Guru Bhandari, Amara Naseer, and Leon Moonen. 2021. CVEfixes: automated collection of vulnerabilities and their fixes from open-source software. In Proceedings of the 17th International Conference on Predictive Models and Data Analytics in Software Engineering (PROMISE 2021). Association for Computing Machinery, New York, NY, USA, 30–39.
>
> [3] Yangruibo Ding, Yanjun Fu, Omniyyah Ibrahim, Chawin Sitawarin, Xinyun Chen, Basel Alomair, David Wagner, Baishakhi Ray, and Yizheng Chen. 2025. Vulnerability Detection with Code Language Models: How Far Are We? In Proceedings of the IEEE/ACM 47th International Conference on Software Engineering (ICSE '25). IEEE Press, 1729–1741.
>
> [4] Fan, J., Li, Y., Wang, S., & Nguyen, T. N. (2020). A C/C++ code vulnerability dataset with code changes and CVE summaries. In Proceedings of the 17th IEEE/ACM International Conference on Mining Software Repositories (pp. 508–512).
>
> [5] Nikitopoulos, G., Dritsa, K., Louridas, P., & Mitropoulos, D. (2021). CrossVul: A cross-language vulnerability dataset with commit data. In Proceedings of the 29th ACM Joint European Software Engineering Conference and Symposium on the Foundations of Software Engineering (pp. 1565–1569).
>
> [6] Ruan, B., Liu, J., Zhao, W., & Liang, Z. (2024). VulZoo: A comprehensive vulnerability intelligence dataset. In Proceedings of the 39th IEEE/ACM International Conference on Automated Software Engineering (pp. 2334–2337).
>
> [7] Chen, Y., Ding, Z., Alowain, L., Chen, X., & Wagner, D. (2023). DiverseVul: A new vulnerable source code dataset for deep learning based vulnerability detection. In Proceedings of the 26th International Symposium on Research in Attacks, Intrusions and Defenses.
>
> [8] Bhatt, D., et al. (2023). Purple Llama CyberSecEval: A secure coding benchmark for language models. arXiv preprint arXiv:2312.04724.
>
> [9] Bhatt, D., et al. (2024). CyberSecEval 2: A wide-ranging cybersecurity evaluation suite for large language models. arXiv preprint arXiv:2404.13161.
>
> [10] Wan, Z., et al. (2024). CyberSecEval 3: Advancing the evaluation of cybersecurity risks and capabilities in large language models. arXiv preprint arXiv:2408.01605.
>
> [11] Liu, Z., Shi, J., & Buford, J. F. (2024). CyberBench: A multi-task benchmark for evaluating large language models in cybersecurity. In Proceedings of the AAAI 2024 Workshop on Artificial Intelligence for Cyber Security.

---

> ### Author Response · Authors · 2025-11-23
> **Response to Reviewer kj1d (4/6)**
>
> **Q5: Detection and identification phases:**
> *The separate evaluation of vulnerability detection and identification might be misleading. I think all false positives from the detection step should also be considered in the input to the identification step, which should significantly degrade the identification results.*
> *How would unifying the detection and identification pipeline affect the results in Section 5?*
>
> **A5**: The reason for seperating these phases is that we wanted to assess how general purpose LLMs are capable of these tasks individually. In addition to that, the passing of all the false positives from the detection step to the identification step naturally lead the LLM to generating an incorrect CVE (hallucination). Additionally, we conduct an experiment to gauge GPT-4o's performance when both steps combined in Table 2 below. As expected, when non-vulnerable samples flagged as vulnerable (false positives) enter the identification phase, the model predicts significantly more incorrect CVE IDs, which degrades overall identification accuracy compared to the decoupled setting. We have clarified this trade-off and reported the joint detection–identification results in the revised manuscript.
>
> Table 2: CVE identification result using GPT-4o when vulnerability detection and identification evaluation steps are combined.
> | **CVE Subset**            | **Top1 Accuracy** |**Top5 Accuracy** |
> |--------------------------|-----------------|-|
> |True Positives        | 53.77        |  73.58 |
> |False Positives        | 34.09        |  52.27 |
> |Overall |48.00|67.33|
>
>
> **Q6: Vulnerability details and scope**
>
> *Q6.1: Did you only focus on the vulnerabilities in the Linux kernel project, or do you include any vulnerabilities that affect Linux systems?*
>
> **A6.1**: We selected CVEs associated with the Linux platform, not just the kernel.
>
> *Q6.2: Related to the above point, why did you only focus on Linux kernel vulnerabilities? How does this affect the applicability of your dataset?*
>
> **A6.2**: We focused on Linux because it is a widely adopted operating system across servers, mobile devices, and personal computers. In addition, the Linux ecosystem provides publicly available discussions, patch code, and commit messages for vulnerabilities.
>
> The resulting dataset can be applied to several downstream tasks, such as  building assistants for security analysts, training LLM-based vulnerability triage tools and powering automated mitigation systems.
>
>
> **Q7: Details of the RAG system**
> *Could you elaborate more on the developed RAG system used to create the dataset? What was the knowledge base items? what were the queries used?*
>
> **A7**: The knowledge base of the RAG system is the generated portfolios of 2000 CVEs. The queries and design choices used for the RAG system are presented in Appendix E of the paper.
>
> "
> The major components in the RAG system are as follows:
> • Dataset Chunking: converting large text corpus into different segments.
> • Embeddings Generation: convert the textual data into embeddings for similarity and faster retrieval.
> • Similarity Search: Identifying the similar chunks that are related to a specific query.
> • Large Language Model: This uses the retrieved chunks to generate a response for a given query.
> We have used Langchain (Chase & contributors, 2022) python library for implementing the archi tecture of our RAG system. In specific, we used LangChain’s RecursiveCharectorTextSplitter to break the large amount of data corpus per CVE into smaller chunks for easier processing. These chunks are then converted into OpenAI embeddings for better representation in Vector space and are stored using Facebook’s Similarity search for vectors (FAISS) technique (Johnson et al., 2019). To select the relevant chunks from this vector store, we have used Maximal Marginal Relevance (Carbonell & Goldstein, 1998) to select chunks that are relevant to the query and also are different from each other. This helps us in minimizing the selection of any duplicate data from the text corpus. The LLM used in RAG pipeline is GPT-4o for its superior reasoning capabilities at the time of generating the datasets. To further improve determinism, the temperature of the LLM is fixed at 0, eliminating unintended randomness in the generated responses.
> "
>
> In addition to that, we have also provided the system prompt and a RAG prompt to extract one attribute (out of 23) in Appendix E.2 and E.3.

---

> ### Author Response · Authors · 2025-11-23
> **Response to Reviewer kj1d (5/6)**
>
> **Q8: Benign code and imbalanced datasets**
> *What was the source for benign code in your dataset? How would having a more balanced (benign vs vulnerable ratio) dataset affect the results in Section 5?*
>
> **A8**: The benign code used in our vulnerability detection evaluation is extracted from the patch commit diffs on GitHub contained in our CVE portfolios. In Section 5, we can reliably obtain the patched (benign) code on 83 out of the 200 CVEs, which is why the original test dataset is imbalanced. To study the impact of class imbalance, we constructed a balanced test set by retaining only CVEs that have both vulnerable and benign code (83 vulnerable and 83 non-vulnerable samples) and discarding CVEs without a corresponding patch. The results on this balanced dataset are reported in Table 3 below, where all LLMs still exhibit poor detection performance. Since our main conclusions are driven by the models’ inability to reliably identify the vulnerable class, making the dataset more balanced does not change our findings about the limitations of current LLMs for vulnerability detection.
>
> Table 3: Vulnerability detection results on a balanced subset (83 vulnerable vs. 83 non-vulnerable samples).
> | **Model**            | **Precision** |**Recall** |**F1** |
> |--------------------------|-----------------|-|-|
> |GPT-4o        | 48.79        |  48.80 |  48.78
> |Claude 4        | 48.35            | 48.80    | 45.11|
> |Gemini 2.5        | 28.72            | 45.18   | 32.04 |
> |Qwen 3      | 51.82 |  51.81  |  51.70
> |Llama 3.3        | 58.90  |  55.43  | 38.78

---

> ### Author Response · Authors · 2025-11-24
> **Updates to our manuscript**
>
> We have updated our manuscript according to our rebuttal. The detailed changes are as follows:
>
> Q1: Section 1 and Appendix A
>
> Q2: Section 5 & Appendix P
>
> Q4: Appendix A
>
> Q5: Section 6
>
> Q8: Appendix N

---

### Official Review · Reviewer_fiHy · 2025-10-29

**Soundness:** 2
**Presentation:** 3
**Contribution:** 2
**Rating:** 4
**Confidence:** 4

**Summary:**

The paper introduces DIAGVULN, a new multi-turn, conversational benchmark for evaluating the vulnerability assessment capabilities of LLMs.  The authors identify critical limitations in existing benchmarks, such as a narrow focus on single data sources, a lack of contextual information (e.g., root cause, exploit details), and a preference for single-turn tasks.  To address this, DIAGVULN is constructed by aggregating data from 13 diverse sources into a comprehensive "Vulnerability Portfolio".  A Retrieval-Augmented Generation (RAG) system is then used to generate 46,000 QA pairs across 23 categories for 2,000 CVEs.  A key part of the methodology is a novel validation pipeline that uses an "LLM-as-a-Judge" calibrated with Conformal Prediction based on human annotations for 100 CVEs, which provides a scalable way to ensure the quality of the RAG-generated answers.  The authors use DIAGVULN to benchmark five SOTA LLMs, finding that while they perform moderately on surface-level information extraction, they substantially fail at tasks requiring deeper reasoning, such as mitigation, exploit explanation, and patch analysis.

**Strengths:**

1. The paper addresses a critical and timely need. As security teams are overwhelmed and LLMs are being integrating into security workflows, a benchmark to measure their true assessment capabilities is urgently required. The paper correctly identifies the flaws of existing benchmarks (single-source, single-turn).

2. The paper's strongest methodological contribution is the validation pipeline. Manually labeling 46,000 QA pairs is infeasible. The proposed solution—using LLM-as-a-Judge and then statistically calibrating this judge using Conformal Prediction (CP) from a small is an elegant and sound approach to building a high-confidence synthetic dataset.
It provides a robust guarantee on the quality of the RAG generation process.

3. The "Vulnerability Portfolio" concept, which aggregates 13 diverse structured and unstructured sources (from NVD to mailing lists), is a major strength. It provides the "holistic" context that is missing from other benchmarks and necessary for real-world analysis.

4. The 3-step evaluation procedure (Detection $\rightarrow$ Identification $\rightarrow$ QA) effectively mimics a practical analyst workflow, making the benchmark far more realistic than simple, single-shot QA datasets.

**Weaknesses:**

1. This overclaiming of "ground truth" taints the main experiment in Section 5. The benchmark evaluates SOTA LLMs (using web search) by comparing their answers against the RAG-generated answers (from the static portfolio). A low "Correctness" score (Table 6) implies the SOTA LLM is wrong. However, it could simply be a disagreement between the SOTA LLM's retrieval (web search) and the paper's RAG system. The live web search might even be more correct or up-to-date. The paper fails to address this ambiguity, fundamentally weakening the evaluation claims. The benchmark measures deviation from the DIAGVULN-RAG system, not necessarily deviation from objective truth.

2. The paper claims “first conversational benchmark” yet Ruan et al. (VulZoo, ASE’24) already pair CVEs with summaries, exploits and patches; the only novelty is turning these into 23 QA templates.

3. The evaluation pits LLMs against each other, not against (i) classical vulnerability detectors (CodeQL, Clang SA, CPAChecker) or (ii) smaller BERT-style models fine-tuned on existing corpora.

4. Limited Scope of Validation: The human validation (used to calibrate the CP) checked 100 CVEs. While the CP framework provides statistical guarantees, the paper isn't fully transparent about the diversity of these 100 CVEs or how representative they are of the 2,000 CVEs in the final QA set.

**Questions:**

1. My primary concern is the "ground truth" claim. Would you be willing to re-frame this? The RAG-generated answers are a high-quality synthetic baseline, not "ground truth." How do you account for the possibility, in your Section 5 evaluation, that a SOTA LLM's answer (from a web search) is correct and your RAG-generated answer (from a static portfolio) is incorrect or outdated? A disagreement does not automatically mean the SOTA LLM failed.

2. The paper aggregates scores across 23 attributes but never shows concrete failure cases. Could you provide a concrete failure analysis, for example, list 10 CVEs where all five LLMs score <3 on Mitigation, and explain the common pattern (missing context, ambiguous patch, etc).

---

> ### Author Response · Authors · 2025-11-22
> **Response to Reviewer fiHy (Part 1/3)**
>
> We thank the reviewer for the detailed and clear comments on our work. Through this rebuttal, we intend to answer the questions and make clarifications.
>
> **Q1: Clarification of "ground truth".**
> *This overclaiming of "ground truth" taints the main experiment in Section 5. The benchmark evaluates SOTA LLMs (using web search) by comparing their answers against the RAG-generated answers (from the static portfolio). A low "Correctness" score (Table 6) implies the SOTA LLM is wrong. However, it could simply be a disagreement between the SOTA LLM's retrieval (web search) and the paper's RAG system. The live web search might even be more correct or up-to-date. The paper fails to address this ambiguity, fundamentally weakening the evaluation claims. The benchmark measures deviation from the DIAGVULN-RAG system, not necessarily deviation from objective truth.*
> *My primary concern is the "ground truth" claim. Would you be willing to re-frame this? The RAG-generated answers are a high-quality synthetic baseline, not "ground truth."*
>
> **A1**:
>
> - **Correctness of RAG.** We manually verified 100 CVEs with domain-expert human evaluators and found that the RAG’s accuracy, shown in Table 1 below, is very close to our target level (Correctness score of 5) for almost all CVE attributes. The only notable exception is Patch Code, which attains a lower average Correctness score of 3.98, reflecting the higher difficulty of this attribute.
>
> - **Ground Truth.** We should call the 100 initial human verified CVEs as ground truth dataset, while calling the RAG-generated 2000-CVE dataset as the "RAG-generated soft-labeled dataset". We will clarify this in the revised manuscript.
>
> Table 1: Human-annotated Correctness scores averaged over 100 human assesment samples for 23 RAG-generated CVE attributes.
>
> | **Attribute**            | **Correctness** |     | **Attribute**         | **Correctness** |     | **Attribute**       | **Correctness** |
> |--------------------------|-----------------|-----|-----------------------|-----------------|-----|---------------------|-----------------|
> | Function Name            | 4.88            |     | Exploit Explanation   | 4.96            |     | Crash Dump          | 4.84            |
> | File Name                | 4.96            |     | Exploit Code          | 4.36            |     | Patch Explanation   | 4.92            |
> | Vulnerable OS Components | 5.00            |     | Exploited Versions    | 4.93            |     | Patch Date          | 4.96            |
> | Weakness Type            | 4.88            |     | Abusable Interfaces   | 4.80            |     | Patched Versions    | 4.96            |
> | Root Cause               | 5.00            |     | Exploit Steps         | 5.00            |     | Code Difference     | 4.87            |
> | Secure Coding Violations | 4.92            |     | Privileges Required   | 4.84            |     | Patch Code          | 3.98            |
> | Mitigation               | 4.86            |     | Exploit Privileges    | 4.96            |     | Score Explanations  | 4.99            |
> | CIA Impact               | 4.71            |     | Remote Exploitability | 5.00            |     |                     |                 |

---

> ### Author Response · Authors · 2025-11-22
> **Response to Reviewer fiHy (Part 2/3)**
>
> **Q2: Comparison to Vulzoo.**
> *The paper claims “first conversational benchmark” yet Ruan et al. (VulZoo, ASE’24) already pair CVEs with summaries, exploits and patches; the only novelty is turning these into 23 QA templates.*
>
> **A2:** While VulZoo provides a useful aggregation of raw vulnerability records, our work differs in both the scope and the depth of processing. Specifically,
>
> 1. Our dataset performs not only simple aggregation but also extraction, processing, and semantic interpretation of each vulnerability record. VulZoo mainly compiles raw information and does not expand or contextualize the data.
> 2. VulZoo does not organize data into well-defined dimensions that capture key aspects of vulnerabilities. Our dataset introduces 6 direct attributes (e.g., patch code) and 17 additional structured attributes (e.g., exploit availability, root cause category, patch metadata, affected subsystem, etc.), enabling deeper reasoning and downstream analyses.
> 3. Our dataset also integrates 8 crucial vulnerability sources that VulZoo does not cover. For example, we include Red Hat advisories, as well as developer-focused security forums such as Seclists and Bugzilla, which contain rich discussions, patch rationale, and impact details that are missing from VulZoo.
> 4. In our work, in addition to a *scalable* end-to-end pipeline for generating data, we also present a *LLM-as-a-judge* to automatically evaluate our data samples with confidence using conformal prediction.
>
> **Our contributions:**
>
> Our main contribution is that, beyond extracting and collecting the data, we 1) curate the raw data into detailed portfolios of each CVE across all attributes, and 2) use the portfolios to generate ready-to-use QA pairs that can directly support downstream LLM applications, such as building assistants for security analysts (e.g., an LLM agent that explains root causes or summarizes exploitability), training LLM-based vulnerability triage tools (e.g., models that classify severity, detect duplicates, or map issues to affected components), and powering automated mitigation systems (e.g., agents that propose patch strategies or compare alternative fixes).
>
> As pointed by *reviewer h3gR*, our dataset makes valuable contributions: "It is a valuable benchmark for evaluating the possible extent of use of LLMs in critical cybersecurity situations and the fine-grained questions provide more insight than previous benchmarks".
>
> **Q3: LLMs and classical vulnerability detection techniques.**
> *The evaluation pits LLMs against each other, not against (i) classical vulnerability detectors (CodeQL, Clang SA, CPAChecker) or (ii) smaller BERT-style models fine-tuned on existing corpora.*
>
> **A3:** Our goal in this work is not to benchmark against classical vulnerability detection tools, but to evaluate the capability of modern LLMs in end-to-end vulnerability assessment. This includes not only detection, but also reasoning about root causes, understanding exploitability, and generating remediation suggestions, where traditional static analyzers (e.g., CodeQL, Clang SA, CPAchecker) and fine-tuned BERT-style models are not designed to provide. For this reason, we compare LLMs against each other to surface differences in their security reasoning ability, rather than against static analyzers whose goals and output formats are fundamentally different.

---

> ### Author Response · Authors · 2025-11-22
> **Response to Reviewer fiHy (Part 3/3)**
>
> **Q4: Diversity of 100 CVEs for ground truth.**
> *Limited Scope of Validation: The human validation (used to calibrate the CP) checked 100 CVEs. While the CP framework provides statistical guarantees, the paper isn't fully transparent about the diversity of these 100 CVEs or how representative they are of the 2,000 CVEs in the final QA set.*
>
> **A4**: The chosen 100 CVEs for ground truth are picked randomly from the available set of CVEs based on rich contextual information across attributes. Choosing these CVEs also helps in validating the RAG system to its full potential across different attributes. For instance, the year distribution of these CVEs is as follows:
>
> | year:count | year:count | year:count | year:count |
> |---------|----------|-----------|-----------|
> | 2007: 2 | 2009: 1 | 2010: 6  | 2011: 11 |
> | 2013: 16 | 2014: 5 | 2015: 1  | 2016: 6  |
> | 2017: 4 | 2018: 5 | 2019: 9  | 2020: 12 |
> | 2021: 10 | 2022: 9 | 2023: 2 | 2024: 1  |
>
> The 100 CVEs also fall across diverse OS components as follows:
>
> | OS Component | Count |
> |------------------------------------|-------|
> | Networking                         | 30    |
> | Filesystems & Storage              | 18    |
> | Memory Management                  | 5     |
> | Core Kernel                        | 6     |
> | Virtualization (KVM / Xen / vhost) | 8     |
> | IPC & Local Communication          | 2     |
> | Security & Namespaces              | 6     |
> | Display & Console (TTY / FBdev)    | 4     |
> | Device Drivers (SPI / USB / misc)  | 3     |
> | Input Devices                      | 1     |
> | Sound Subsystem                    | 2     |
> | User-space Tools / Applications    | 2     |
> | Unknown / Not Specified            | 13    |
>
>
> **Q5: Concrete failure cases**
> *The paper aggregates scores across 23 attributes but never shows concrete failure cases. Could you provide a concrete failure analysis, for example, list 10 CVEs where all five LLMs score <3 on Mitigation, and explain the common pattern (missing context, ambiguous patch, etc).*
>
> **A5**: After analyzing the mitigation results generated by SOTA LLMs for 12 CVEs, we have identified the following limitations across all LLMs:
> 1. **Obvious mitigation strategy**: A well known strategy to fix any vulnerability is to update them. In all the cases, the LLMs tend to directly suggested to a version that is not vulnerable. In some cases where there is information that a specific Linux version is vulnerable, the only provided solution is to upgrade to a version greater than that.
> 2. **Absence of mitigation explanation**: In one specific case "CVE-2014-7283" where the vulnerability is present in ```xfs``` file system that allows users to crate directories and cause hash map corruption, the LLMs bluntly suggested upgrading to a version that does not have this issue rather than suggesting a tool ```xfs_repair``` that could directly fix this vulnerability.
> 3. **Incorrect details and Hallucinations**: The SOTA LLMs have shown hallucinations in Linux versions, distro-specific guidances to fix vulnerability, and mitigation explanation. Out of 12 CVEs, LLMs have hallucinated across 4 CVEs. In CVE-2020-27152, it is observed that all 5 models hallucinated the Linux version that is fixed to upgrade.
> 4. **Generic security guidelines:** Instead of presenting the mitigation detail that a security researcher / programmer wants, the LLMs tend to just start recommending boilerplate steps such as monitoring logs, limiting user access, deploying intrusion detection systems, and so on. While these suggestions are helpful in a wider scope, these might not work in a specific use case of fixing / mitigating a CVE.
>
> The CVE IDs that have the least mitigation scores and used for this experiment are as follows:
> |           |             |           |                  |
> |---------------|---------------|----------------|----------------|
> | CVE-2013-6431 | CVE-2014-7283 | CVE-2019-15031 | CVE-2019-3701 |
> | CVE-2020-12659 | CVE-2020-12769 | CVE-2020-25211 | CVE-2020-27152 |
> | CVE-2021-45868 | cve-2022-28356 | CVE-2023-40791 | CVE-2023-4921 |
>
>
> We have added them in the updated manuscript.

---

> ### Author Response · Authors · 2025-11-24
> **Updates to our manuscript**
>
> We have updated our manuscript according to our rebuttal. The detailed changes are as follows:
>
> Q1 - Section 4 “soft labels”
>
> Q4 - Appendix I
>
> Q5 - Section 6 & Appendix K

---

### Official Review · Reviewer_qNHz · 2025-10-31

**Soundness:** 3
**Presentation:** 2
**Contribution:** 3
**Rating:** 4
**Confidence:** 3

**Summary:**

The paper introduces DiagVuln, a novel, large-scale conversational benchmark dataset to evaluate the SOTA LLMs on vulnerability assessment. The author aimed to address the gaps in the existing vulnerability detection datasets by curating data from 13 structured and non-structured sources, and developed “Vulnerability Portfolio” with rich context information on the Linux Kernel CVEs. The dataset contains 2000 CVEs across 23 question-answer categories, including detection, localisation, classification, root cause analysis, exploit reasoning, impact assessment, and patch. The key contribution of the work is an automated framework that generates QA pairs from the vulnerability portfolio using RAG. The authors used an LLM-as-a-judge framework with conformal prediction to validate the quality of the automatically generated data. Finally, the paper benchmarks five SOTA LLMs on a subset of the dataset, and the results show that LLMs struggle with complex, reasoning-heavy vulnerability analysis tasks.

**Strengths:**

- Collection of holistic vulnerable data from diverse data sources (structured and unstructured) and curating a large-scale scale rich contextful “Vulnerability Portfolio”, which might also be used for future fine-tuning tasks.
- The data collection pipeline is scalable with RAG for QA generation and LLM-as-a-judge for validation.
- The results demonstrate the limitations of the current LLMs and provide a future research roadmap for software security.

**Weaknesses:**

- The CVE Attribution method for the unstructured data solely depends on the regular expressions to extract CVEs. This approach assumes that the unstructured developer discussion and code comments are correctly tagged with a complete single CVE ID, which may not be true in the real-world developers' settings. There might not be a mention of CVE IDs, or the CVE references can be partial, or even the discussion may contain multiple CVE references. How do the authors address this issue?
- No mention of the number of samples in the held-out calibration set for the conformal prediction. How did the authors ensure the calibration set size was sufficiently large to represent the whole dataset?
- The authors provided a detailed process of human assessment with the required time on 100 QA pairs. However, no quantitative data on this assessment are provided. How to compare the performance of LLM-as-a-Judge against the human annotators?
- In the evaluation section, the authors have mentioned that the 200 CVE subset was selected based on the “richest contextual information”. Do the authors consider the potential data leakage issue in this case?  How will the model perform against the less documented vulnerabilities?
- LLMs have poor capabilities of reasoning about vulnerabilities. Using LLMs to curate vulnerability questions/answers may generate low quality data

**Questions:**

- What steps were taken in case of missing, ambiguous CVE IDs for the unstructured sources beyond the regular expression? How much does this affect the portfolio curation of less documented vulnerabilities?

- The "Correctness" of the RAG-generated answers is validated against the source portfolio. How does the pipeline evaluate for scenarios where the source data itself is incorrect or incomplete?

- Is the evaluation result also generalizable to the less documented vulnerabilities?

- Among 6,342 CVEs, how many are used to build RAG database, how the RAG database is built?

---

> ### Author Response · Authors · 2025-11-22
> **Response to Reviewer qNHz (Part 1/4)**
>
> We thank the reviewer for the detailed and clear comments on our work. Through this rebuttal, we intend to answer the questions and make clarifications.
>
>
> **Q1.1: Missing or ambiguous CVEs in unstructured datasets.**
>  *The CVE Attribution method for the unstructured data solely depends on the regular expressions to extract CVEs. This approach assumes that the unstructured developer discussion and code comments are correctly tagged with a complete single CVE ID, which may not be true in the real-world developers' settings. There might not be a mention of CVE IDs, or the CVE references can be partial, or even the discussion may contain multiple CVE references. How do the authors address this issue?*
> *What steps were taken in case of missing, ambiguous CVE IDs for the unstructured sources beyond the regular expression?
>
> **A1.1**: We have five unstructured data sources (seclists, lkml, Openwall, Packetstorm, and Redhat). When extracting the data from unstructured sources, we handled the cases mentioned by the reviewer as follows:
>
> - For Redhat, there is a **one-one mapping** between each webpage and the CVE ID based on the URL.
> - For lkml, Openwall, Packetstorm, we start with the NVD CVE ID, and we browse the NVD CVE webpage to check if there is any link pointing to those websites. If so, we map the linked page with the CVE ID, which **guarantees one-one mapping** between CVE ID and the unstructured source, even if those posts contain multiple CVEs.
> - For Seclists, we **did not observe any case** where a post contains multiple CVE ID when using the regular expressions to extract CVE ID from the seclists posts.
>
> In summary, we did not encounter any ambiguity case when mapping unstructured data to the corresponding CVE.
>
> **Q1.2 Less documented vulnerabilities.**
> *How much does this affect the portfolio curation of less documented vulnerabilities?*
>
> **A1.2**: When only a few documents are found regarding a specific CVE, that leads to empty fields in the portfolio. This leads to a "Not found" placeholder in the QA pairs. We clarified this in the revised version of the manuscript.
>
> **Q2: Size of data used for conformal prediction.**
> *No mention of the number of samples in the held-out calibration set for the conformal prediction. How did the authors ensure the calibration set size was sufficiently large to represent the whole dataset?*
>
> **A2**: Following prior works [1][2], we use 100 samples that are annotated by domain experts as the held-out calibration set, which is mentioned in lines 87-88 on page 2 in our manuscript. Using a calibration size on the order of 100 data samples for 90% coverage is common for conformal prediction. For example, [1] showed that even modest calibration sizes yield well-controlled error rates for outlier detection, and [2] utilized 100 calibration points to construct accurate 90% prediction intervals in both regression and image classification. Additionally, we sample the 100 calibration samples uniformly at random from the full dataset so that they form an i.i.d. subsample from the same distribution. This allows us to rely on the standard finite-sample conformal coverage guarantee while keeping annotation costs manageable.
>
> [1] Bates, Stephen, et al. "Testing for outliers with conformal p-values." The Annals of Statistics 51.1 (2023): 149-178.
>
> [2] Plassier, Vincent, et al. "Efficient conformal prediction under data heterogeneity." International Conference on Artificial Intelligence and Statistics. PMLR, 2024.

---

> > ### Author Response · Authors · 2025-11-22
> > **Response to Reviewer qNHz (Part 2/4)**
> >
> > **Q3: LLM judge vs. human annotators.**
> > *The authors provided a detailed process of human assessment with the required time on 100 QA pairs. However, no quantitative data on this assessment are provided. How to compare the performance of LLM-as-a-Judge against the human annotators?*
> >
> > **A3**: We showcase the human-assigned Faithfulness, Correctness, and Completeness for 23 CVE attributes averaged over 100 samples selected for human assessment in Table 1 below. Additionally, in Figure 3 of our paper, we have presented the conformal calibrated scores of 200 RAG-generated QA pairs. Specifically, we plot the average LLM-assigned Faithfulness, Correctness, and Completeness per attribute, where square markers (■) denote perfect agreement (zero deviation) with human scores, and triangles (▲) denote non-zero LLM–human deviations with 90% conformal confidence intervals as error bars. Here, LLM–human deviation is defined as the absolute difference between human and LLM scores, and the 90% confidence intervals are calibrated on the 100 human-annotated samples, as mentioned in Section 3.3 in our manuscript. Figure 3 highlights that the RAG-generated attributes achieve high evaluation scores and exhibit strong LLM–human agreement under 90% conformal prediction confidence.
> >
> > Table 1: Human-annotated scores averaged over 100 human assessment samples for 23 RAG-generated CVE attributes.
> >
> > | **Attribute**            | **Faithfulness** | **Correctness** | **Completeness** |
> > |--------------------------|------------------|-----------------|------------------|
> > | Function Name            | 4.88             | 4.88            | 4.80             |
> > | File Name                | 4.96             | 4.96            | 4.94             |
> > | Vulnerable OS Components | 5.00             | 5.00            | 4.99             |
> > | Weakness Type            | 4.88             | 4.88            | 4.87             |
> > | Root Cause               | 5.00             | 5.00            | 5.00             |
> > | Secure Coding Violations | 4.92             | 4.92            | 4.96             |
> > | Mitigation               | 4.86             | 4.86            | 4.92             |
> > | CIA Impact               | 4.70             | 4.71            | 4.88             |
> > | Exploit Explanation      | 4.96             | 4.96            | 4.95             |
> > | Exploit Code             | 4.40             | 4.36            | 4.40             |
> > | Exploited Versions       | 4.98             | 4.93            | 4.99             |
> > | Abusable Interfaces      | 4.80             | 4.80            | 4.80             |
> > | Exploit Steps            | 5.00             | 5.00            | 4.98             |
> > | Privileges Required      | 4.84             | 4.84            | 4.82             |
> > | Exploit Privileges       | 4.96             | 4.96            | 4.94             |
> > | Remote Exploitability    | 5.00             | 5.00            | 5.00             |
> > | Crash Dump               | 4.84             | 4.84            | 4.81             |
> > | Patch Explanation        | 4.92             | 4.92            | 4.92             |
> > | Patch Date               | 4.96             | 4.96            | 4.96             |
> > | Patched Versions         | 4.96             | 4.96            | 4.95             |
> > | Code Difference          | 4.88             | 4.87            | 4.82             |
> > | Patch Code               | 4.40             | 3.98            | 4.04             |
> > | Score Explanations       | 4.99             | 4.99            | 5.00             |

---

> ### Author Response · Authors · 2025-11-22
> **Response to Reviewer qNHz (Part 3/4)**
>
> **Q4: Data leakage issue.**
> *In the evaluation section, the authors have mentioned that the 200 CVE subset was selected based on the “richest contextual information”. Do the authors consider the potential data leakage issue in this case? How will the model perform against the less documented vulnerabilities?*
>
> **A4**: From the experimental results on CVE Identification (Table 5 in our manuscript), we demonstrated that most LLMs in the evaluation have very limited specialized knowledge about most of the 200 CVE subset. For instance, GPT-4o identifies only 10 correct CVE IDs out of 200 samples without searching for additional online information. Additionally, we conducted experiments to investigate the data leakage issue by evaluating GPT-4o on CVE attribute question-answering (QA) with 10 less documented CVEs. The results in Table 2 below show that the Correctness scores slightly decrease on 16/23 QA attributes. This reduction in performance can be explained either by the weaker documentation footprint of these CVEs or by potential leakage on the original 200 CVEs with richer context. However, given the poor CVE identification performance and the modest performance reduction in attribute-level QA on less-documented CVEs, any data leakage, if present, appears minor. We will also discuss the potential data leakage issue in our limitation.
>
> Table 2: CVE attribute question-answering evaluation with GPT-4o using 10 less documented CVEs. Here, **boldface** indicates where less documented CVEs have lower Correctness score for a CVE attribute.
> | **Attribute**            | 200 Rich-Context CVEs | 10 Less-Documented CVEs |
> |--------------------------|-----------------|------------------------|
> | Function Name            | 3.67            | **2.20**                   |
> | File Name                | 3.85            | **3.80**                   |
> | Vulnerable OS Components | 3.96            | 4.30                   |
> | Weakness Type            | 3.02            | **2.40**                   |
> | Root Cause               | 4.22            | **2.56**                   |
> | Secure Coding Violations | 3.57            | **3.40**                   |
> | Mitigation               | 2.11            | 2.40                   |
> | CIA Impact               | 3.92            | **3.10**                   |
> | Exploit Explanation      | 2.69            | **2.30**                   |
> | Exploit Code             | 3.70            | **3.40**                   |
> | Exploited Versions       | 2.94            | **2.50**                   |
> | Abusable Interfaces      | 3.61            | **3.20**                   |
> | Exploit Steps            | 3.40            | **2.80**                   |
> | Privileges Required      | 3.77            | 3.80                   |
> | Exploit Privileges       | 3.03            | 3.90                   |
> | Remote Exploitability    | 4.39            | **4.00**                   |
> | Crash Dump               | 3.09            | **2.00**                   |
> | Patch Explanation        | 3.50            | **3.44**                   |
> | Patch Date               | 1.31            | 1.70                   |
> | Patched Versions         | 1.84            | 2.60                   |
> | Code Difference          | 2.53            | **1.80**                   |
> | Patch Code               | 1.50            | **1.40**                   |
> | Score Explanations       | 2.78            | 2.90                   |

---

> ### Author Response · Authors · 2025-11-22
> **Response to Reviewer qNHz (Part 4/4)**
>
> **Q5: LLMs for vulnerability reasoning.**
> *LLMs have poor capabilities of reasoning about vulnerabilities. Using LLMs to curate vulnerability questions/answers may generate low quality data*
>
> **A5**: We are **not** using the LLM as a reasoning solution. Instead, we are using LLMs to **retrieve vulnerability information**. The explanations of the vulnerabilities (e.g., exploit and patch explanations) are extracted by RAG system based on factual CVE documents. We have clarified this in the updated manuscript.
>
> **Q6: Correctness of source portfolio.**
> *The "Correctness" of the RAG-generated answers is validated against the source portfolio. How does the pipeline evaluate for scenarios where the source data itself is incorrect or incomplete?*
>
> **A6**: We have selected **authentic and trustworthy websites** that are reputed among the community for vulnerability analysis. These websites are used by other dataset curation papers [3, 4, 5]. We have added some clarification regarding this in the manuscript.
>
> [3] Chen, Y., Ding, Z., Alowain, L., Chen, X., and Wagner, D. 2023. DiverseVul: A new vulnerable source code dataset for deep learning–based vulnerability detection. Proceedings of the 26th International Symposium on Research in Attacks, Intrusions and Defenses (RAID ’23). ACM, New York, NY, USA, 654–668.
>
> [4] Nikitopoulos, G., Dritsa, K., Louridas, P., and Mitropoulos, D. 2021. CrossVul: A cross-language vulnerability dataset with commit data. Proceedings of the 29th ACM Joint Meeting on European Software Engineering Conference and Symposium on the Foundations of Software Engineering (ESEC/FSE 2021). ACM, New York, NY, USA, 1565–1569.
>
> [5] Sun, J., Chen, J., Xing, Z., Lu, Q., Xu, X., and Zhu, L. 2024. Where is it? Tracing the vulnerability-relevant files from vulnerability reports. Proceedings of the IEEE/ACM 46th International Conference on Software Engineering (ICSE 2024). IEEE/ACM, 1–13.
>
> **Q7: RAG database.**
> *Among 6,342 CVEs, how many are used to build RAG database, how the RAG database is built?*
>
> **A7**: Out of the 6,342 total portfolio queries, we have used 2,000 CVEs for the RAG dataset. These 2,000 CVEs were selected because they best represent the entries that contain the most complete information across all attributes. The RAG system details are presented in Appendix E of the paper:
>
> "The major components in the RAG system are as follows:
> - Dataset Chunking: converting large text corpus into different segments.
> - Embeddings Generation: convert the textual data into embeddings for similarity and faster retrieval.
> - Similarity Search: Identifying the similar chunks that are related to a specific query.
> - Large Language Model: This uses the retrieved chunks to generate a response for a given query.
>
> We have used Langchain (Chase & contributors, 2022) python library for implementing the architecture of our RAG system. In specific, we used LangChain’s RecursiveCharectorTextSplitter to break the large amount of data corpus per CVE into smaller chunks for easier processing. These chunks are then converted into OpenAI embeddings for better representation in Vector space and are stored using Facebook’s Similarity search for vectors (FAISS) technique (Johnson et al., 2019). To select the relevant chunks from this vector store, we have used Maximal Marginal Relevance (Carbonell & Goldstein, 1998) to select chunks that are relevant to the query and also are different from each other. This helps us in minimizing the selection of any duplicate data from the text corpus. The LLM used in RAG pipeline is GPT-4o for its superior reasoning capabilities at the time of generating the datasets. To further improve determinism, the temperature of the LLM is fixed at 0, eliminating unintended randomness in the generated responses."
>
> In addition to that, we have also provided the system prompt and a sample RAG prompt to extract one attribute (out of 23) in Appendix E.2 and E.3.

---

> ### Author Response · Authors · 2025-11-24
> **Updates to our manuscript**
>
> We have updated our manuscript according to our rebuttal. The detailed changes are as follows:
>
> Q1.2 - Section 3.1 & Section 3.2
>
> Q3 - Appendix S
>
> Q4 - Section 6 & Appendix Q
>
> Q5 - Section 3.2
>
> Q6 - Section 1

---

### Official Review · Reviewer_h3gR · 2025-10-31

**Soundness:** 2
**Presentation:** 3
**Contribution:** 2
**Rating:** 4
**Confidence:** 3

**Summary:**

The paper introduces a conversational benchmark about CVEs from multiple from multiple data sources to evaluate the capabilities of LLMs as cybersecurity assistants in three key areas (vulnerability detection, CVE identification and Q&A about the identified vulnerability). It also introduces a dataset curation and validation pipeline to extend the benchmark. The evaluation on the benchmark with a combination of LLM-as-a-judge and conformal prediction shows significant gaps in current LLMs

**Strengths:**

•	It is a valuable benchmark for evaluating the possible extent of use of LLMs in critical cybersecurity situations and the fine-grained questions provide more insight than previous benchmarks.


•	In addition to LLM-as-a-judge, the authors use conformal prediction techniques to establish high-confidence bounds on the acceptable difference between human and LLM labels.

**Weaknesses:**

•	In Section 5 Prompting Strategy, the authors mention a model-agnostic system prompt, but recent research (https://aclanthology.org/2025.naacl-long.73/, https://arxiv.org/abs/2408.11865) has shown that LLMs are very sensitive to prompt construction. As a result, the assumption that a single prompt will work equally well across all the tested models might not hold true. It will be interesting to see how optimized prompts can work for each LLM, specially the open-weights ones.

•	The authors do not use structured generation for the model responses even though all the models are capable of structured generation, either through vLLM or their own API. This can reduce the error rates significantly.

•	While the authors mention that the dataset can be used to fine-tune domain-specific LLMs, it introduces benchmark contamination and any results on the benchmark by a fine-tuned LLM will not be admissible. To do that successfully needs a privately held-out test set.

**Questions:**

1.	Why was structured generation skipped? It makes for a more structured evaluation as well as improving the capabilities of less-sophisticated models

2.	Do you have any plans for a test-set that can be used to evaluate fine-tuned models in the future? Specifically, it would be helpful to have the test set from sources other than the train set.

3.	Were any other models evaluated for RAG besides GPT-4o? Since this model is used for both the RAG pipeline and the LLM-as-a-judge, the scores cannot be compared similarly for other LLMs

---

> ### Author Response · Authors · 2025-11-22
> **Response to Reviewer h3gR (Part 1/2)**
>
> We thank the reviewer for the detailed and clear comments on our work. Through this rebuttal, we intend to answer the questions and make clarifications.
>
>
> **Q1: Different prompt for different LLMs.**
> *In Section 5 Prompting Strategy, the authors mention a model-agnostic system prompt, but recent research (https://aclanthology.org/2025.naacl-long.73/, https://arxiv.org/abs/2408.11865) has shown that LLMs are very sensitive to prompt construction. As a result, the assumption that a single prompt will work equally well across all the tested models might not hold true. It will be interesting to see how optimized prompts can work for each LLM, specially the open-weights ones.*
>
> **A1**: We have conducted additional experiments using model-specific optimized prompts for Claude 4 and Llama 3.3 for the CVE attribute question-answering (QA) evaluation. The results are summarized in Table 1 below. From this table, we can observe that using optimized prompts can improve the Correctness scores on 12/23 QA attributes for Claude 4, and 14/23 QA attributes for Llama 3.3, while other attributes exhibit similar or slightly worse performance. The reason for this is that our model-agnostic prompt for this evaluation already incorporates several prompt-sensitivity mitigation strategies suggested by [1], including enforcing strict answer formats and providing step-by-step instructions for technical attributes (e.g., Exploit Code, Patch Code). To further optimize the prompts, we instruct Claude 4 to answer questions directly and concisely to limit over-explanation and speculation [2], and impose stricter context restrictions and structured reasoning instructions on Llama 3.3 to reduce hallucination [3]. However, the results in Table 1 show that **the optimized prompts do not yield drastic gain over our model-agnostic version, suggesting that our evaluation in Section 5 is robust to reasonable prompt variants**. We have included our prompt-sensitivity mitigation strategies in the revised manuscript, and added the per-LLM optimized-prompt results in the Appendix of the updated manuscript.
>
> Table 1: CVE attribute question-answering evaluation using optimized prompts for Claude 4 and Llama 3.3. Here, **boldface** indicates where the optimized prompt for each LLM improves the Correctness score for a CVE attribute.
> |**Attribute**| **Claude 4 - Model-Agnostic** | **Claude 4 - Optimized** | **Llama 3.3 - Model-Agnostic** | **Llama 3.3 - Optimized** |
> |-|-|-|-|-|
> | Function Name| 3.16 | 2.93| 3.74| **3.78**|
> | File Name| 3.53| 3.12 | 3.78 | **3.79** |
> | Vulnerable OS Components | 4.50 | **4.63**  | 4.07 | **4.38** |
> | Weakness Type | 3.30 | **3.72** | 3.31     | **3.53**       |
> | Root Cause    | 4.65    | 4.50          | 4.42     | 4.42     |
> | Secure Coding Violations | 4.06  | 3.96   | 3.10 | 2.69  |
> | Mitigation | 2.05  | **3.04** | 1.91 | **2.28**  |
> | CIA Impact | 4.32  | 3.56   | 4.12 | 2.89  |
> | Exploit Explanation  | 3.00  | **3.33** | 2.77 | **2.84**  |
> | Exploit Code | 3.02  | **3.67** | 3.62 | **3.80**  |
> | Exploited Versions | 2.59  | 3.03   | 2.57 | **2.76**  |
> | Abusable Interfaces  | 3.06  | **3.88** | 3.19 | **3.40**  |
> | Exploit Steps  | 3.51  | 3.21   | 3.28 | 3.24  |
> | Privileges Required  | 4.33  | 4.00 | 3.49 | 3.32  |
> | Exploit Privileges | 4.34  | **4.41**   | 2.65 | **4.48**  |
> | Remote Exploitability  | 4.56  | 3.04   | 4.48 | 3.88  |
> | Crash Dump | 1.74  | **3.90** | 3.88 | 3.88  |
> | Patch Explanation  | 3.89  | 2.25   | 3.38 | 2.28  |
> | Patch Date | 1.52  | **2.50** | 1.88 | **2.30**  |
> | Patched Versions   | 1.79  | **2.14** | 1.64 | **2.26**  |
> | Code Difference  | 1.83  | **2.73** | 2.50 | **2.76**  |
> | Patch Code | 2.54  | 1.41   | 1.28 | **1.42**  |
> | Score Explanations | 3.79  | **3.94** | 3.25 | 3.18  |
>
>
>
> [1] Anagnostidis, Sotiris, and Jannis Bulian. "How susceptible are llms to influence in prompts?." arXiv preprint arXiv:2408.11865 (2024).
>
> [2] Anthropic. Reduce hallucinations. Claude Documentation (2024).
>
> [3] Meta AI. Prompt engineering / Prompting Llama models. Llama Documentation (2024).
>
>
> **Q2: Structured generation.**
> *The authors do not use structured generation for the model responses even though all the models are capable of structured generation, either through vLLM or their own API. This can reduce the error rates significantly.*
> *Why was structured generation skipped? It makes for a more structured evaluation as well as improving the capabilities of less-sophisticated models.*
>
> **A2**: Since we only input one question at a time into the LLM, which will output one answer corresponding to only that question, structured generation is not necessary in our evaluation.

---

> ### Author Response · Authors · 2025-11-22
> **Response to Reviewer h3gR (Part 2/2)**
>
> **Q3: Held out test set.**
> *While the authors mention that the dataset can be used to fine-tune domain-specific LLMs, it introduces benchmark contamination and any results on the benchmark by a fine-tuned LLM will not be admissible. To do that successfully needs a privately held-out test set.*
> *Do you have any plans for a test-set that can be used to evaluate fine-tuned models in the future? Specifically, it would be helpful to have the test set from sources other than the train set.*
>
> **A3**: We will split the collected QA pairs into training and test sets. In total, we have created the QA pairs for 2,000 CVEs across 23 attributes. This data set can be split into an 80/20 split of 1,600 CVEs in the training dataset and 400 CVEs in the testing dataset.
>
>
>
>
> **Q4: RAG with different models.**
> *Were any other models evaluated for RAG besides GPT-4o? Since this model is used for both the RAG pipeline and the LLM-as-a-judge, the scores cannot be compared similarly for other LLMs.*
>
> **A4**: In addition to GPT-4o, other LLMs such as Gemini 1.5 Pro and Llama 3 were also tested when we performed the evaluation. GPT-4o performed the best, so we chose it for our experiment. Below is an example of an exploit explanation for a CVE "CVE-2002-2438", which shows that GPT-4o gives the details of the exploit clearly across different dimensions:
>
> - GPT-4o
>
>         TCP firewalls could be circumvented by
>         sending SYN packets with other flags,
>         such as the RST flag, set. These packets
>         were not correctly discarded by the
>         Linux TCP stack after firewalling,
>         allowing the attacker to bypass firewall
>         rules. Additionally, SYN+FIN attacks
>         were reported to bring Linux machines
>         to their limits, indicating a potential
>         denial-of-service condition. The
>         protection added in Linux 2.4.20
>         involved checking for the RST flag when
>         a SYN packet is received, which was the
>         main attack scenario.
>
> - Llama 3
>
>         TCP firewalls could be circumvented by
>         sending a SYN packet with other flags
>         (like e.g. FIN flag) set, which was not
>         correctly discarded by the Linux TCP
>         stack after firewalling.
>
> - Gemini 1.5 Pro
>
>         Not found.
>
> Another crucial reason for selecting GPT-4o is for its thinking capabilities. Any other models can be selected which possess similar or better thinking capabilities.

---

> ### Author Response · Authors · 2025-11-24
> **updates to our manuscript**
>
> We have updated our manuscript according to our rebuttal. The detailed changes are as follows:
>
> Q1 - Section 6 & Appendix O
>
> Q3 - Section 4.1
>
> Q4 - Section 3.2 & Appendix F.1

---

### Author Response · Authors · 2025-11-30
**Summary of responses**

Dear Area Chair,

We are writing to provide a consolidated summary of the changes and additions made following the reviewer comments, including revisions to the manuscript and all newly conducted experiments. We believe the rebuttal and revision have addressed all questions and concerns from the reviewers.

For Reviewer h3gR, we have answered the following questions.
- The size of the held-out test split that can be used from the dataset is clarified in the manuscript which helps in further utilizing this dataset for downstream tasks.
- We have presented detailed elaboration of the RAG system regarding the choice of LLM used, and usage of 2000 CVEs as RAG database size in Appendix F.
- We have also clarified the reason for not using structured generation in our evaluation.

For Reviewer qNHz, we have answered the following questions.
- We have detailed the cases of how we handled less documented and missing CVE information while aggregating all the vulnerability information.
- We have updated the scores of human annotators and LLM-as-Judge for generating the soft labels in Appendix S.
- We have added additional citations that support the correctness of our selected vulnerability sources used to compile the dataset.

For Reviewer fiHy, we have answered the following questions.
- More detailed motivations and comparisons to the prior vulnerability datasets were added in Appendix A.
- The diversity represented by the 100 CVEs used for creating initial ground truth is clearly detailed in Appendix I.
- An observed pattern and analysis of SOTA LLMs on CVEs is added as a discussion in Appendix K.

For Reviewer kj1d, we have answered the following questions.
- We have included a more detailed explanation about the contribution and motivations of our work in Appendix A.
- We clarified the clear scope and applicability of our dataset and listed the downstream tasks that can be achieved using this dataset.
- A detailed rationale behind the choice of using a RAG system instead of LLMs for vulnerability reasoning is presented in Section 3.2.


**New experiments and analysis**
Alongside these clarifications, we have incorporated the following additional experiments into the manuscript:

(1) Evaluation of model-specific prompt adjustments for SOTA LLMs, which did not exhibit significant improvement. This is added in Appendix O.

(2) In-context learning performance assessment of an LLM using few-shot samples drawn from our dataset, and we observed an increase in overall performance, demonstrating the utility and potential of our dataset. The results are presented in Appendix P.

(3) Evaluation that combines the vulnerability detection and identification phases, which showed that the combined evaluation on SOTA LLMs led to increased hallucinations. The results are presented in Appendix R.

(4) Evaluation of SOTA LLMs on less-documented CVEs, which did not yield a significant drop of performance. The results are presented in Appendix Q.

(5) Evaluation using a balanced vulnerability dataset, which proved that LLMs show poor performance even on a balanced dataset, thus validating our original results. The results are presented in Appendix N.

---

### Note · Program_Chairs · 2026-01-17
**Submission Desk Rejected by Program Chairs**

The following references in this submission do not refer to real documents and/or have major errors in bibliographic information:

 Daniel H Robinson and Kevin R Leonard. The likert scale: An underused and misunderstood tool in evaluation. Performance Improvement, 58(4):10-17, 2019.